# Compensatory action of different types of *cis*-regulatory elements buffers the striped expression of *Drosophila* pair-rule genes

Matthew D. Fischer[1,2,*], Kristen Au[1,2], Minh Lê[1], Patricia Graham[1] and Leslie Pick[1,2,‡]

## ABSTRACT

The striped expression of pair-rule genes in *Drosophila* embryos is a paradigm for understanding transcriptional control of development. Pair-rule striped expression is regulated by two types of *cis*-regulatory elements: stripe-specific elements respond to non-periodic cues in different regions of the embryo to establish individual stripes while 7-stripe elements simultaneously regulate all stripes, responding to pair-rule genes expressed in stripes. Here, we assess roles of stripe-specific versus 7-stripe elements for the pair-rule gene *ftz*. We show that loss of a *ftz* stripe 2 element is compensated by 7-stripe elements, even though they respond to different spatiotemporal cues. We next investigate whether similar rules apply to the classic *eve* stripe2 element. Animals homozygous for a genomic deletion of *eve* stripe2 are viable and fertile; stripe 2 expression is perturbed early but re-establishes sufficiently to regulate downstream target genes. However, temperature or genetic stress decrease viability of *ftz* and *eve* stripe 2 deletion mutants. Thus, these stripe-specific elements contribute to robustness but are not absolutely required for segment formation. Two separate routes to establishing stripes, stripe-specific and 7-stripe elements, buffer each other, adding complexity to embryonic patterning.

KEY WORDS: Enhancers, *Drosophila* development, Pair-rule genes, *ftz*, *eve*

## INTRODUCTION

The expression of regulatory genes in distinct regions of the early embryo establishes unique identities within a field of undifferentiated cells. The *Drosophila* blastoderm represents an ideal model in which to study this process as sets of regulatory genes are expressed sequentially to establish the segmented body plan of the fly. Pivotal to this process are the pair-rule genes (PRGs), the first genes to be expressed in a repetitive pattern along the anterior-posterior body axis. For example, *fushi tarazu* (*ftz*) and *even-skipped* (*eve*) are expressed in complementary patterns of seven stripes along the anterior-posterior axis in the primordia of body regions missing in

[1]Department of Entomology, University of Maryland, College Park, MD 20742, USA. [2]Molecular & Cell Biology Graduate Program, University of Maryland, College Park, MD 20742, USA.
*Present address: Center for Computational and Genomic Medicine, Children's Hospital of Philadelphia Research Institute, Philadelphia, PA 19104, USA.

‡Author for correspondence (lpick@umd.edu)

L.P., 0000-0002-4505-5107

corresponding pair-rule (PR)-mutant embryos (Hafen et al., 1984; Carroll and Scott, 1985; Macdonald et al., 1986; Harding et al., 1986; Frasch and Levine, 1987; Lawrence and Johnston, 1989) (see Fig. S1). Since these genes are required for segment establishment, they are expressed before segments are morphologically visible. Striped expression patterns are short-lived, arising and then rapidly fading during an ∼2.5 h period after fertilization. Reporter gene analysis in transgenic embryos has been used widely to identify *cis*-regulatory elements (CREs) and transcription factors that control these patterns (e.g. Hiromi et al., 1985; Hiromi and Gehring, 1987; Dearolf et al., 1989; Pankratz et al., 1990; Goto et al., 1989; Arnosti et al., 1996; Harding et al., 1989; Pick et al., 1990; Jiang et al., 1991; Small et al., 1991, 1992, 1996; Gutjahr et al., 1994; Fujioka et al., 1995, 1999, Baltruk et al., 2022; Butler et al., 1992; Klingler and Gergen, 1993).

Early studies of PRG expression suggested that so-called primary PRGs, such as *eve* and *hairy*, are controlled directly by maternal and gap proteins, whereas the 7-stripe patterns of so-called secondary PRGs, such as *ftz*, are controlled solely by pre-positioned primary PRG products (Ingham and Martinez-Arias, 1986; Howard and Ingham, 1986; Carroll et al., 1988; Ingham and Gergen, 1988; Ingham et al., 1988; Baumgartner and Noll, 1990; Howard and Struhl, 1990) (reviewed by Akam, 1987; Lawrence, 1992; Hartman et al., 1994; Langeland et al., 1994). Reporter gene studies supported these notions. Stripe-specific CREs that direct expression of individual PR stripes were identified for *hairy* (Howard et al., 1988; Riddihough and Ish-Horowicz, 1991). Analysis of the control of *eve* stripes using reporter transgenes identified stripe-specific CREs, the dissection of which established a paradigm for how non-periodic information provided by maternal and gap proteins is translated into the periodic, striped expression of PRGs in stripes. The work on *eve* stripe CREs, and in particular on the *eve* stripe 2 CRE, had a large impact because it answered a compelling question, combining genetic and biochemical approaches to demonstrate that *eve* stripe 2 is activated in a broad region by Bcd and Hb but repressed at the anterior and posterior borders by gap proteins Gt and Kr, to generate precise stripe boundaries (Goto et al., 1989; Stanojevic et al., 1989; Treisman and Desplan, 1989; Warrior and Levine, 1990; Stanojevic et al., 1991; Small et al., 1991, 1992; Arnosti et al., 1996; Fujioka et al., 1999; Vincent et al., 2018; Lopez-Rivera et al., 2020). This is now a standard textbook example of how a body plan can be patterned through transcriptional regulation (e.g. Gilbert, 2010; Hartwell et al., 2017). The *eve* locus also has stripe-specific CREs for stripes 3 and 7 (Yu and Small, 2008; Struffi et al., 2011), stripe 1, stripe 5, and stripes 4 and 6 (Small et al., 1996; Fujioka et al., 1999; Sackerson et al., 1999; Yu and Small, 2008; Struffi et al., 2011). Although some details remain to be worked out, sets of broadly expressed activators along with gap repressors appear to position each *eve* stripe. A late-acting (L) 7-stripe CRE was also identified for *eve* (Goto et al., 1989; Harding et al., 1989; Jiang et al., 1991; Manoukian and Krause, 1992; Fujioka et al., 1995, 2002).

*DEVELOPMENT*

In contrast, for *ftz*, a 6 kb upstream region was shown to contain two separate 7-stripe CREs: the zebra element (Z) and the upstream element (UPS) (Hiromi et al., 1985; Hiromi and Gehring, 1987; see Fig. S2 purple boxes). Dissection of the Z (Dearolf et al., 1989) and of the UPS (Pick et al., 1990) identified short regions of each CRE responsible for activation or repression, but stripe-specific subregions were not identified in either. The UPS mediates autoregulation via direct binding of Ftz and Ftz-F1 (Yu et al., 1997), suggesting that initial activation in 7-stripes directed by Z is increased by the UPS. This view was bolstered by earlier findings that *ftz* expression was changed in mutants for several other PRGs (Howard and Ingham, 1986; Ingham and Gergen, 1988; Carroll and Scott, 1986), thus explaining establishment of the periodic *ftz* pattern by PR protein(s) pre-positioned to impact all seven stripes in a coordinated manner. However, a careful analysis of *ftz* expression (Yu and Pick, 1995) and identification of candidate stripe-specific CREs at the extreme ends of a genomic region sufficient to rescue *ftz* mutants (Calhoun and Levine, 2003; Pick et al., 1990) indicated additional complexity.

How different are the modes of regulation of *eve* and *ftz*, or, more generally, primary versus secondary PRGs? A landmark study identified stripe-specific CREs for *ftz* and other PRGs (Schroeder et al., 2011). *odd-skipped* (*odd*) and *runt* (*run*) contain stripe-specific elements in addition to a 7-stripe CRE. *sloppy-paired* (*slp1* and *slp2*) and *paired* (*prd*) are regulated in a stripe-specific fashion for anterior stripes, but only 7-stripe CREs have been identified for the remaining stripes (Gutjahr et al., 1993, 1994; Schroeder et al., 2004; Prazak et al., 2010; Hang and Gergen, 2017; Ochoa-Espinosa et al., 2005). Finally, only stripe-specific CREs (no 7-stripe CRE) have been identified for *hairy* (*h*; *hry*) (Howard et al., 1988; Hooper et al., 1989; Howard and Struhl, 1990; Riddihough and Ish-Horowicz, 1991). This led to a re-classification of primary and secondary PRGs with *eve*, *h*, *ftz*, *odd* and *run* as primary PRGs – having CREs that respond directly to maternal and gap input – and *slp* and *prd* as secondary PRGs, with their stripes established by other PRGs (except for anterior stripes). In summary, the striped expression patterns of *eve*, *ftz*, *odd* and *runt*, although initially thought to be regulated by different mechanisms, are controlled by a combination of stripe-specific and 7-stripe CREs. In general, the 7-stripe CREs appear to direct expression later in development than the stripe-specific CREs (Goto et al., 1989; Harding et al., 1989; Jiang et al., 1991; Zhao et al., 2023; Birnie et al., 2023; Fischer et al., 2024).

Recently, our lab demonstrated that, for *ftz*, neither 7-stripe CRE is absolutely required for formation of the *ftz* 7-stripe expression pattern. Deletion of Z only significantly impacted the expression of one stripe – stripe 4, for which no stripe-specific CRE has been identified – with defects in the corresponding segments of larvae and adults (Graham et al., 2021). This suggested that a threshold of *ftz* transcription is necessary to direct segment formation. Other experiments showed that this threshold must be met during the transition of stage 5 (cellular blastoderm) into stage 6 (gastrulation) (Birnie et al., 2023; Zhao et al., 2023). Deletion of the UPS confirmed that it is required only for late *ftz* expression after gastrulation and showed that it is not required for correct expression of downstream target genes (Fischer et al., 2024).

Here, we investigated whether stripe-specific CREs play a fundamental role in *ftz* patterning. We found that deletion of the *ftz* stripe 2 CRE impacted early expression of stripe 2, but embryos recovered a full 7-stripe pattern, expressed downstream target genes in a wild-type fashion, and were viable and fertile. We examined whether stripe-specific elements are dispensable for other PRGs by deleting the *eve* stripe 2 CRE (eve2) from the genome. This led to

early loss of stripe 2 but, as for the *ftz* stripe 2 CRE deletion, expression recovered. Although homozygous viable and fertile, both stripe2 CRE deletions resulted in decreased survival compared to wild type. Further, neither *ftzΔ2* nor *eveΔ2* deletion mutants were viable in combination with *ftz* or *eve* null alleles, respectively, although transheterozygous escapers were observed with weak alleles. These results suggest that *eve* and *ftz* use similar modes of regulation to establish and maintain stripes and that the stripe 2-specific CREs for both genes contribute to robustness. They further demonstrate that different types of CREs, which act over different time ranges, can compensate for each other despite being regulated by different sets of transcription factors.

## RESULTS

### A stripe-specific CRE directs early *ftz* stripe 2 expression

Prior research identified partially overlapping genomic fragments sufficient for directing transgene expression in a single stripe in the region presumed to correspond to *ftz* stripe 2 (Fig. 1A). The larger fragment, ftz2+7 [originally named ftz_(−6)] was identified through computational analysis of genomic regions in which maternal and gap protein binding sites clustered and directs expression in the regions presumably corresponding to *ftz* stripes 2 and 7 (Schroeder et al., 2004, 2011). The core region, ftz2 (originally named *ftz413*) was identified through random 3′ exonuclease digestion of the UPS (Pick et al., 1990) and is nested within ftz2+7 (Fig. 1A; Fig. S2; Table S1) (Pick et al., 1990). Genomic regions corresponding to ftz2 and ftz2+7 were amplified by PCR and inserted into the same site of the *pattB-lacZ* reporter gene plasmid and the same attP landing site of the genome using the phiC31 system (Venken et al., 2006, 2011). Clear expression of the *ftz2>lacZ* transgene was detected in stripe 2 (Fig. 1B). In the reverse orientation, expression in stripe 2 was substantially stronger and extended over the surrounding parasegments (Fig. 1C), consistent with older data (Pick et al., 1990). Expression of the *ftz2+7F>lacZ* transgene was detected faintly in stripe 2 (Fig. 1E). In the reverse orientation, expression

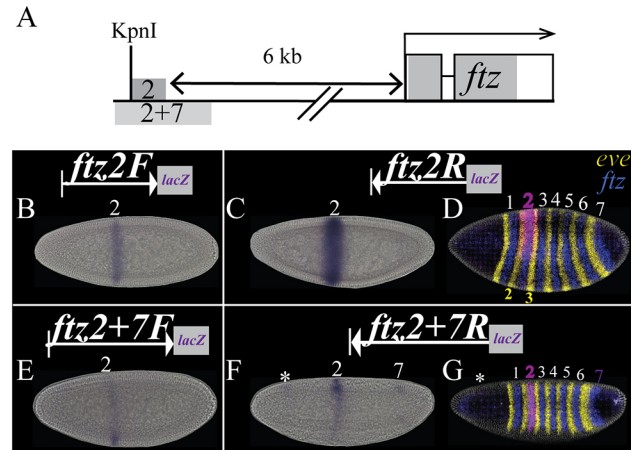

**Fig. 1. Transgenic reporters of ftz2 and ftz2+7 co-express with endogenous *ftz* stripe 2.** (A) Schematic of the *ftz* locus. We renamed the 413 (Pick et al., 1990) and ftz_(−6) (Schroeder et al., 2011) CREs to ftz2 and ftz2+7, respectively. (B,C,E,F) Digoxygenin-labeled probes for chromogenic *in situ* hybridization using a *lacZ* probe of embryos containing transgenes for ftz2 or ftz2+7 in forward (F) or reverse (R) orientation. (D,G) Fluorescence confocal images of HCR *in situ* in the indicated transgenic reporter lines: *eve*, yellow; *ftz*, blue; *lacZ*, purple; Hoechst 34580, gray. Histogram adjusted in ImageJ to enhance qualitative analysis of co-expression domains. White numbers, *ftz* stripes; purple numbers, *lacZ* co-expression; yellow numbers, *eve* stripes with detectable *lacZ* co-expression.

was observed in stripe 2 and faintly in stripe 7 (Fig. 1F), similar to previous findings (Schroeder et al., 2011).

Although these stripe-specific elements were thought to recapitulate the expression of endogenous *ftz* stripes, this had not been demonstrated. Therefore, we conducted multiplexed fluorescence *in situ* hybridization chain reaction (HCR) experiments to compare expression of the *lacZ* transgenes to endogenous *ftz* stripes (Fig. 1D,G). This confirmed that *ftz2R>lacZ* expression was detected over the endogenous *ftz* stripe 2, extending into the surrounding *eve* stripe 2 and stripe 3 domains (Fig. 1D) while *ftz2+7R>lacZ* was expressed in a strong stripe overlapping *ftz* stripe 2 and a weak stripe overlapping *ftz* stripe 7 (Fig. 1G) (Schroeder et al., 2011). The weaker expression of stripe 2 for ftz2+7 compared to ftz2 (compare Fig. 1C to 1F) suggests that the 413 bp ftz2 core CRE contains the information necessary for activation of stripe 2 expression while the flanking regions present in ftz2+7 bind repressors of stripe 2, similar to the situation for eve2 (Lopez-Rivera et al., 2020). The expression of all other presumed *ftz* stripe-specific CREs overlapped with the expected *ftz* stripes (Fig. S3; Table S1). Orientation dependency of downstream *ftz* CRE reporter transgenes suggests that a small genomic fragment between the 3′end of *ftz* and the ftz1+5 CRE inhibits transgene expression, similar to insulators and tethering elements reported elsewhere in the genome (Fig. S3E-J) (Batut et al., 2022).

### The *ftz* stripe 2-specific CRE is dispensable for segment development

To examine the role of *ftz* stripe-specific CREs in endogenous *ftz* expression, a precise deletion of ftz2 was generated with CRISPR/Cas9-mediated genome editing (*ftzΔ2*; Materials and Methods; Figs S2, S4, Table S1). To test whether the regions flanking ftz2 have a repressive effect on expression of stripe 2 and/or are required for stripe 7 (Fig. 1B,C versus 1E,F), a second deletion was generated that removes just these flanking regions while retaining the ftz2 CRE (*ftz2ΔFlanking*; Materials and Methods; Figs S2, S5, Table S1). Both deletion mutants were homozygous viable, fertile and have been maintained for many generations as homozygotes. *ftzΔ2* and *ftz2ΔFlanking* adults developed with wild-type features including all body segments (Fig. 2A-C). However, the *ftz2ΔFlanking* females appeared to have larger abdomens than wild type (data not shown).

Larval cuticles were examined to determine whether the wild type-like adult homozygotes were escapers while other homozygotes with severe defects died during development. The overwhelming majority of *ftzΔ2* and *ftz2ΔFlanking* larvae displayed wild-type segmentation (Fig. 2E, 98.9%, n=172/173; Fig. 2G, 99.4%, n=311/313). Although one or two individuals from these deletion mutants showed minor defects (Fig. 2F,H), these proportions are identical to those observed in wild-type controls (Fig. 2D, 0.5%; n=1/173).

### Stripe-specific CREs of *ftz* are insufficient for viability

Given that neither the UPS nor the Z was independently required for *ftz* stripe formation or segment development, we examined whether the combined input of these 7-stripe CREs is necessary for segmentation or if stripe-specific CREs are sufficient to establish segmentation without any 7-stripe CRE contribution. To this end, a *ftzΔUPS^{7S}ΔZ* double mutant was generated by deleting the Z from the homozygous viable *ftzΔUPS^{7S}* background (Figs S2, S6, Table S1). In the absence of both the UPS and Z CREs, the only contributions to *ftz* should be from the stripe-specific CREs.

Unlike any of the other CRE deletion mutants examined, the *ftzΔUPS^{7S}ΔZ* allele is homozygous lethal; self-crosses of

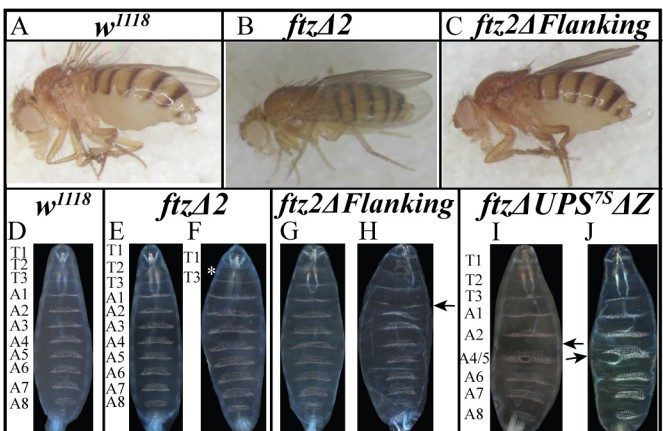

**Fig. 2. The *ftz* stripe 2 CRE is dispensable for segmentation.**
(A-C) Brightfield images of wild-type control (A), *ftzΔ2* (B) and *ftz2ΔFlanking* (C) adults. Wild type, *ftzΔ2* and *ftz2ΔFlanking* develop with all segments. (D-J) Darkfield microscopy of cuticle preparations. (D) Wild type. (E) 98.9% of *ftzΔ2* larvae had wild type-like cuticles. (F) One *ftzΔ2* larva was missing T2 (asterisk). (G) 99.4% of *ftz2ΔFlanking* larvae had wild type-like cuticles. (H) One *ftz2ΔFlanking* larva had a gap in A1 (arrow). (I) Most *ftzΔUPS^{7S}ΔZ* larvae were missing A3 and A4 /A5 were fused (arrow). (J) Some *ftzΔUPS^{7S}ΔZ* larvae were missing A3 and had partial A4/A5 fusion/deletions (arrow).

*ftzΔUPS^{7S}ΔZ/TM3,Sb* did not yield any adult homozygotes. Cuticle preparations revealed that 73% (n=140/192) were wild type-like and 27% developed with segmental abnormalities (Fig. 2I, J), suggesting that all homozygous *ftzΔUPS^{7S}ΔZ* embryos develop segmental abnormalities. These homozygotes were missing segment A3, corresponding to the region specified by *ftz* stripe 4 (see schematic, Fig. S1). Eighteen percent of embryos scored had severe defects, with complete fusion between two other segments, usually A4 and A5 (Fig. 2I), while the other 9% showed moderate segmental defects, with incomplete fusion between two other segments (Fig. 2J; for additional examples, see Fig. S8). However, among 192 cuticles scored by one experimenter (P.G.) and 656 scored by another (K.A.), only one complete PR phenotype was observed. This suggests that, for many *ftz* stripes, stripe-specific CREs can provide sufficient levels and spatial refinement of *ftz* for wild type-like function.

### Early activation of *ftz* stripes is perturbed in *ftzΔ2^{-/-}* and *ftz2ΔFlanking^{-/-}* embryos

We next examined endogenous *ftz* expression in the CRISPR-generated mutants. We found that the order and patterning of *ftz* stripe activation differed from wild type in the embryos of both *ftzΔ2* and *ftz2ΔFlanking* homozygotes. The progression of *ftz* stripe expression was tracked temporally by following the progression of membrane invagination along the periphery throughout stage 5 (Fig. 3, insets). As previously described (Yu and Pick, 1995; Surkova et al., 2008; Schroeder et al., 2011; Graham et al., 2021; Fischer et al., 2024), *ftz* stripes arose in the following order: stripes 1 and 5 (Fig. 3AA); stripes 2 and 3 (Fig. 3AB), and a combined stripe 6/7 that began on the ventral side and progressed dorsally (Fig. 3AC). Stripes 6 and 7 then split while expression increased in stripes 1, 2 and 5 (Fig. 3AD). Higher abundance of *ftz* transcript was then detected in stripe 3, and stripe 4 expression was detected at low levels (Fig. 3AE). Following this, expression in all stripes increased, but stripes 3 and 4 remained weaker (Fig. 3AF). Stripes 3 and 4 then reached comparable expression levels to other stripes, except stripe 7, which was detected at higher levels (Fig. 3AG).

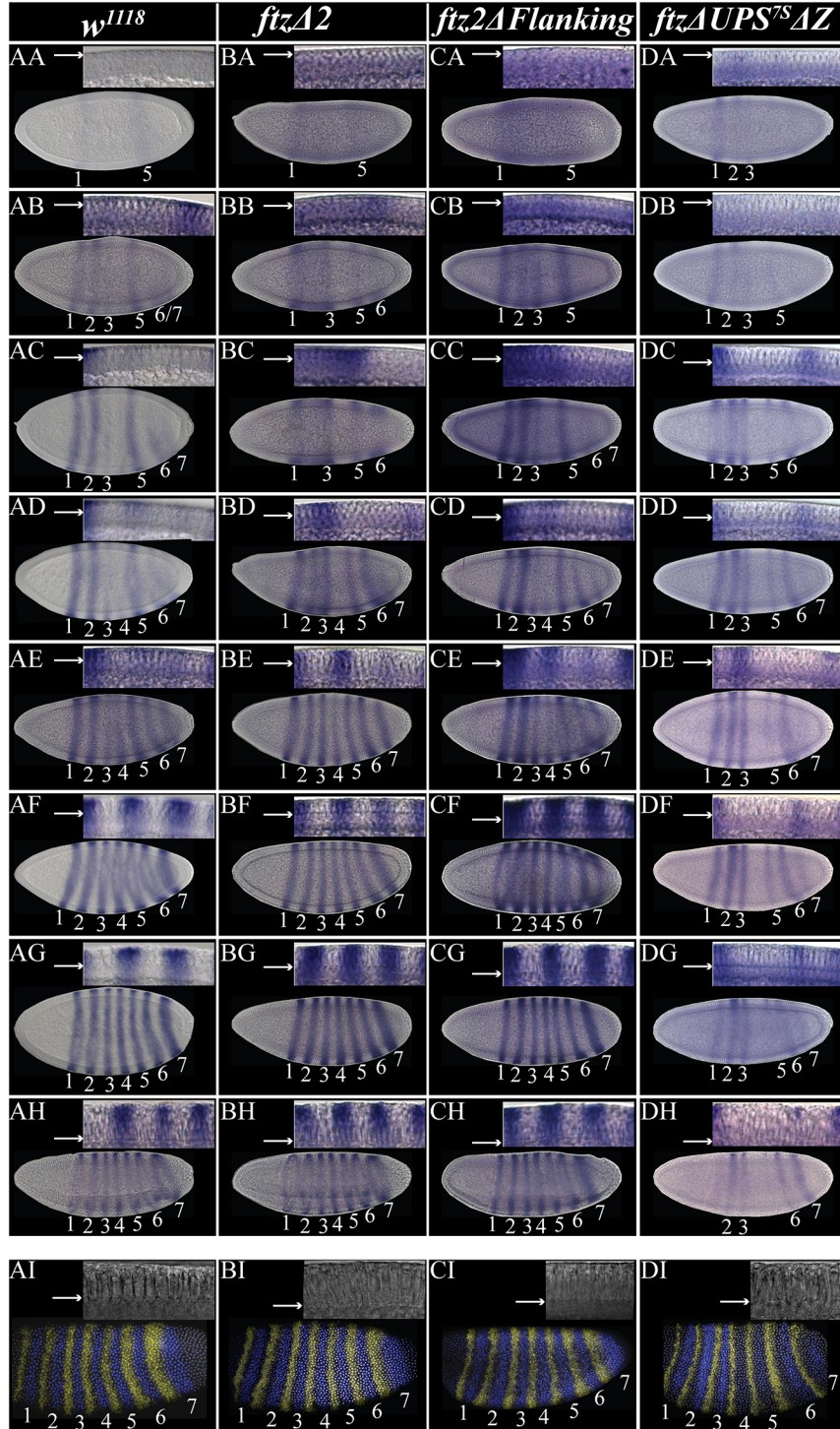

**Fig. 3. The dynamic expression of *ftz* stripe establishment is different in *ftzΔ2*, *ftz2ΔFlanking* and *ftzΔUPS^{7S}ΔZ* mutant embryos.** (AA-DH) Brightfield images of *ftz* expression time series throughout stage 5 (cellular blastoderm) of wild-type (AA-AH), *ftzΔ2* (BA-BH), *ftz2Δ2+7* (CA-CH) and *ftzΔUPS^{7S}ΔZ* (DA-DH) embryos using digoxygenin-labeled probes. Insets in upper-right corner of each panel show the progress of membrane invagination, with arrows indicating the leading edge of the membrane. Embryos are oriented anterior left, dorsal up except BB and BC. (AI,BI,CI,DI) Confocal images of late-stage 5 embryos with HCR using probes targeting *ftz* (blue) and *eve* (yellow). Nuclei stained with Hoechst 34580 (gray).

As the membrane front reached the cortex and cellularization was complete, peak expression was reached for all seven *ftz* stripes (Fig. 3AH).

Three main differences from wild type in *ftzΔ2* embryos were observed. First, a broad band of early expression in the anterior trunk in this mutant (Fig. 3BA-BC). Second, *ftz* stripe 2 took much longer to activate than in wild type (Fig. 3AB in wild type and Fig. 3BE in *ftzΔ2*). Finally, *ftz* stripe 7 activated later and as an individual stripe rather than the broad 6/7 combined stripe observed in wild type (Fig. 3AC versus Fig. 3BE). Despite these differences, the *ftz* pattern in *ftzΔ2* embryos was the same as wild type at peak

expression in the blastoderm (Fig. 3AH,BH). This was validated by quantitation of gene expression levels by HCR, which found that differences in fluorescence intensity and width of *ftz* stripes in wild type, *ftzΔ2* and *ftz2ΔFlanking* late-stage 5 embryos were statistically insignificant (Fig. S7, Table S2). Thus, early perturbations of *ftz* stripe establishment in stripe-specific CRE mutations can be compensated for by 7-stripe CREs.

In *ftz2ΔFlanking*, yet another order of *ftz* stripe expression was observed. First, stripes 6 and 7 were detected as distinct domains (Fig. 3CD,CE), as opposed to a combined stripe in wild type (Fig. 3AC). Second, stripe 2 was broad and more strongly expressed

than in wild type during the majority of the cellular blastoderm (Fig. 3CB-CF). However, it did become refined and restricted to the expected four-cell-wide domain by peak *ftz* expression (Fig. 3CG-CH).

The patterning of *ftz* stripe 2 is the main difference when comparing *ftz∆2* to *ftz2∆Flanking* mutants. While expression within the stripe 2 domain was delayed in *ftz∆2* mutant embryos (Fig. 3BE), the initiation of expression within stripe 2 of *ftz2∆Flanking* mutant embryos was similar to wild type (Fig. 3CB) but refined later than wild type (Fig. 3CC-CG). Additionally, whereas initiation of *ftz* stripe 7 expression began as a broad band that resolved in *ftz∆2* and wild-type embryos (Fig. 3BE), *ftz* stripe 7 was initially very thin and weak in *ftz2∆Flanking* embryos (Fig. 2CE).

Despite these differences of *ftz* dynamics throughout the cellular blastoderm, *ftz* expression in both stripe-specific CRE mutants was indistinguishable from wild type throughout gastrulation and germband extension (Fig. 4A-L). This suggests that the impacts of stripe-specific CREs on *ftz* expression are temporally restricted to early *ftz* expression and their deletion is compensated for by the end of stage 5, likely by the 7-stripe CREs.

### Blastoderm expression of *ftz* is perturbed and rapidly decreases in *ftz∆UPS^{7S}∆Z* embryos

The order of *ftz* stripe establishment differed from wild type in *ftz∆UPS^{7S}∆Z* mutant embryos. Expression of *ftz* was first observed exclusively within the anterior stripe domains, specifically a strong stripe 1 and weak stripes 2 and 3 (Fig. 3DA). Stripe 5 was observed next while stripes 2 and 3 remained faint (Fig. 3DB). Expression of *ftz* in stripes 2, 3 and 5 increased, especially stripe 2 (Fig. 3DC). Simultaneously, stripe 6 began as a thin stripe and without stripe 7. Next, *ftz* expression remained strong in stripes 1, 2 and 3, with stripe 6 becoming clearer, and stripe 5 remaining faint (Fig. 3DD). Faint expression of *ftz* stripe 7 became detectable while stripes 5 and 6 strengthened (Fig. 3DE,DF). At the next time point, stripe 5 and 7 expression increased slightly, though stripe 7 remained faint (Fig. 3DG). From this point forward, *ftz* expression decreased in the stripe 1, 5, 6 and 7 domains, with a substantial decrease in stripe 5 (Fig. 3DH). As the blastoderm fully cellularized, stripes 1 and 5 were nearly undetectable, and stripes 2, 3 and 6 decreased, with stripe 7 remaining strong. As gastrulation began, *ftz* expression was nearly undetectable in stripes 1 and 5, and only dorsal expression was observed for stripes 2, 3, 5 and 6 (Fig. 4M). For the rest of gastrulation, *ftz* expression was undetectable (Fig. 4N-P), likely due to the absence of the UPS, which is necessary for late *ftz* expression (Fischer et al., 2024).

There were several differences during the blastoderm stage between wild type and the *ftz∆UPS^{7S}∆Z* mutant embryos. *ftz* expression began in the anterior of the double mutant, while stripes 1 and 5 formed first in wild type. Next, stripes 6 and 7 formed together in wild type, but were separate in the double mutant, with stripe 7 expression occurring later and at lower levels than in wild type. Third, although *ftz* expression was at peak levels upon the completion of cellularization in wild type, expression had already begun to fade in the double mutant by this point. By the time the ventral furrow started to form, *ftz* expression was nearly undetectable in multiple domains, whereas in wild type, all seven stripes were strong and maintained. Notably, expression of *ftz* was never detected in stripe 4 of the double-mutant embryos, while it was reduced but still detectable in either *ftz∆Z* or *ftz∆UPS^{7S}* homozygotes (Fischer et al., 2024; Graham et al., 2021). These results suggest that: (1) the weak stripe 4 expression observed from the *ftz∆Z^{−/−}* mutants was contributed from the UPS^{7S} CRE and vice versa; (2) the weakness of posterior stripes in the double mutant (e.g. Fig. 3DD) was due to loss of Z; and (3) since stripe 4 is present in *ftz∆Z^{−/−}* mutants but absent in *ftz∆UPS^{7S}∆Z* mutants, the contributions of the UPS CRE to *ftz* expression are more than purely autoregulatory.

### Expression of *slp1* and *en* is perturbed in *ftz∆UPS^{7S}∆Z^{−/−}* embryos

To assess the impact of CRE mutations on Ftz downstream target gene regulation, we examined expression of *slp1*, which is repressed by Ftz, and *engrailed* (*en*), which is activated by Ftz (Howard and Ingham, 1986; DiNardo and O'Farrell, 1987; DiNardo et al., 1988; Carroll et al., 1988; Ingham et al., 1988; Florence et al., 1997; Cadigan et al., 1994; Prazak et al., 2010; Hang and Gergen, 2017; Graham et al., 2021). In wild type, both *slp1* and *en* were expressed in 14 stripes during germband extension (Fig. 5A,F). These expression patterns were indistinguishable from wild type in both *ftz∆2* (Fig. 5B,G) and *ftz2∆Flanking* (Fig. 5C,H) mutant embryos, although we cannot rule out small quantitative differences in expression level using colorimetric *in situ* hybridization. Thus, peak PR expression at the critical point in time – rather than the order and patterning of early PR stripe establishment as mediated by stripe-specific CREs – is most consequential for proper regulation of downstream gene expression and subsequent segmentation of the organism.

In contrast, the *ftz∆UPS^{7S}∆Z* mutant embryos had perturbed downstream expression patterns. *slp1* was expanded in 28% (*n*=23/81) of embryos (∼ 25% homozygotes expected). Fig. 5D shows an embryo in which *slp1* stripes 8 and 9 fused and other stripes, such as 10 and 11, were partially expanded, a pattern observed in 21% of embryos (*n*=17/81); this corresponds to decreases in levels of *ftz* stripes 4 and 5 (Fig. S1). In 7% (*n*=6/81) of embryos, this loss of repression was observed for multiple

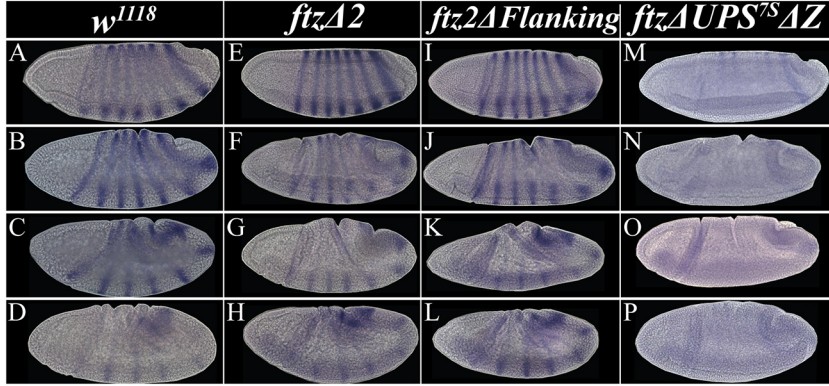

**Fig. 4. *ftz* expression is not maintained throughout gastrulation in *ftz∆UPS^{7S}∆Z^{−/−}* embryos.**
(A-P) Brightfield images of *ftz* expression as a time series throughout gastrulation (stage 6, top row; stage 7, other rows) of wild-type (A-D), *ftz∆2* (E-H), *ftz2∆Flanking* (I-L) and *ftz∆UPS^{7S}∆Z* (M-P) embryos.

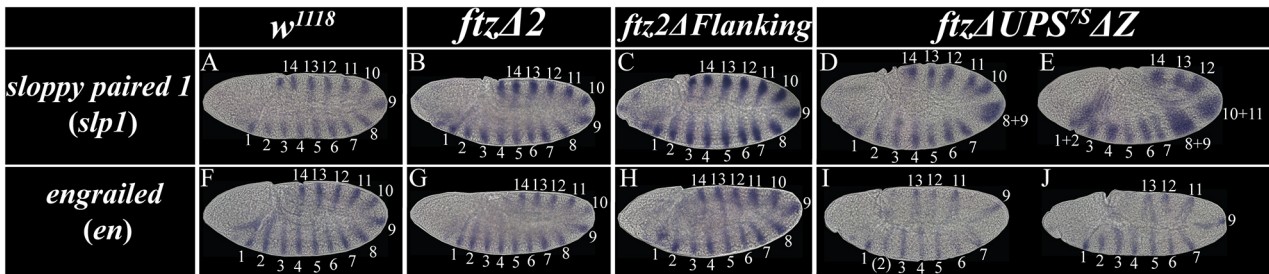

**Fig. 5. Downstream expression of *slp1* and *en* are impacted in parasegments corresponding to reduced *ftz* expression in *ftzΔUPS^{7S}ΔZ* mutant embryos.** (A-J) Brightfield images of *sloppy paired 1* (*slp1*) and *engrailed* (*en*) expression during the germband extension stage of wild-type and mutant embryos. Stripe numbers for each gene are indicated. (A,F) Wild type; (B,G) *ftzΔ2* mutants; (C,H) *ftz2ΔFlanking* mutants; (D,E,I,J) *ftzΔUPS^{7S}ΔZ* mutants.

*slp1* stripes, showing either complete fusion between stripes or slight expansion of individual stripes (Fig. 5E). A similar trend was observed for *en*: 19% (*n*=30/158) of total embryos showed defects in more than one expression domain (Fig. 5I,J), with one single embryo showing loss of all *ftz*-dependent stripes. There were consistent defects in stripes 8, 10 and 14, corresponding to *ftz* stripes 4, 5 and 7. Some embryos had weak *en* expression in other even-numbered parasegments, which are Ftz dependent (Fig. S1).

Connecting the expression changes and segmental phenotypes, this suggests that the level to which *ftz* is reduced in this double mutant is likely below the threshold necessary for proper regulation of downstream segmentation pathway genes in some domains (e.g. *en* stripes 8, 10 and 14, corresponding to *ftz* stripes 4, 5 and 7, respectively) and near the cusp for others (e.g. *en* stripes 2 and 12, corresponding to *ftz* stripes 1 and 6, respectively). These defects correspond to the cuticle phenotypes observed for this mutant (Fig. 2I,J), for which A3 was absent (no *ftz* stripe 4; no *en* stripe 8), and A4/5 were fused together, either partially or completely (due to low *ftz* stripe 5, consistent with perturbations to *en* stripe 10; Fig. S1).

### *eve* and *ftz* stripe-specific CREs play similar roles

Earlier research in the field suggested that *ftz* relies primarily on 7-stripe CREs while *eve* relies primarily on stripe-specific CREs (see Introduction). To test this, we generated an *eve* mutant comparable to *ftzΔ2* by deleting the *eve* stripe 2 CRE (Small et al., 1992) to generate *eveΔ2* (Fig. S9). Surprisingly, *eveΔ2* mutants are homozygous viable, fertile and have been maintained in the lab as a true-breeding stock. No morphological abnormalities were seen in *eveΔ2^{−/−}* adults (Fig. 6A,B). Most larval cuticles similarly displayed wild type-like segmentation (Fig. 6C,D, left). Minor defects in the thorax were observed in some cases (Fig. 6D, middle and right).

To determine how absence of the eve2 CRE impacted *eve* expression, we examined *eve* expression in *eveΔ2* homozygotes (Fig. 7). Insets in upper-right panels show the staging of embryos, based on the progression of membrane invagination for each image. Like *ftz*, establishment of the *eve* stripes occurs in distinct phases, albeit with slight variations within each time class (Surkova et al., 2008). First, there is a single, broad anterior stripe as the nuclei elongate (Fig. 7A). This domain narrows and intensifies into stripe 1

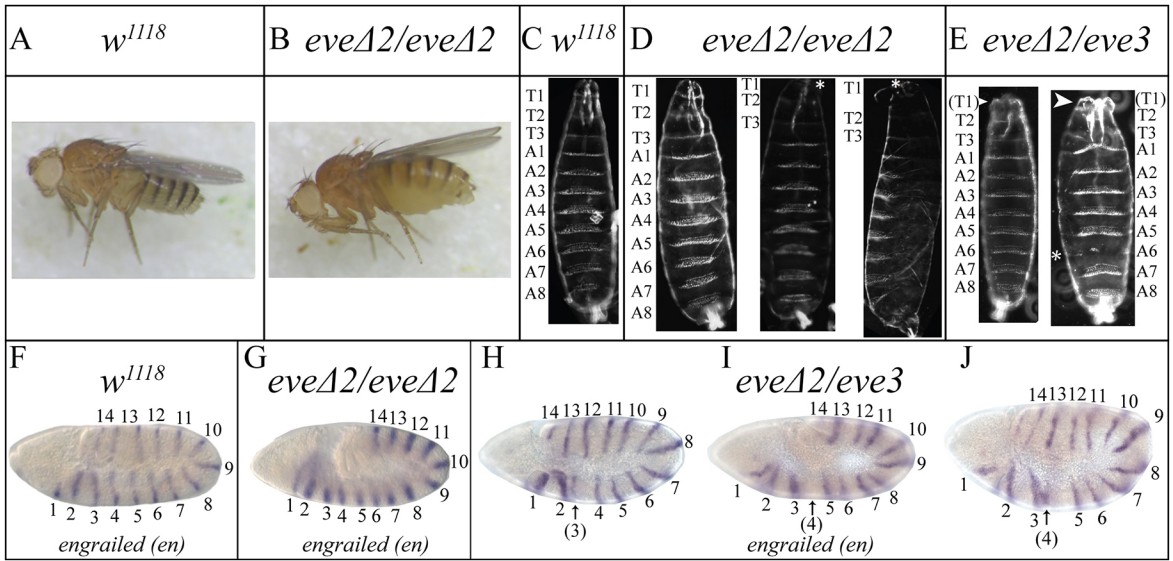

**Fig. 6. The classic *eve* stripe 2 CRE is not required for segmentation.** (A,B) Brightfield images of adults. (B) *eveΔ2* adults are homozygous viable and lack visible segmentation defects. (C-E) Darkfield microscopy images of cuticle preparations. (D, left) Most *eveΔ2* larvae developed with wild type-like cuticles. (D, middle and right) Examples of *eveΔ2* larvae with abnormalities around T1 (asterisks). (E) Larval cuticles of offspring from crosses of *eveΔ2* homozygotes to *eve^{3}/CyO*; left, wild type-like, presumptive *eveΔ2/CyO*; right, anterior defects (arrowhead) and abdominal defect (asterisk), presumptive *eveΔ2/eve^{3}*. (F-J) Downstream gene expression. (F,G) *engrailed* is expressed in 14 stripes. (H-J) *en* expression was altered in presumptive *eveΔ2/eve^{3}* embryos. Numbers refer to *en* stripes. (H) A weak *en* stripe 3 is evident (arrow). (I) A weak *en* stripe 4 is evident (arrow). (J) *eveΔ2/eve^{3}* embryo with stripes 3 and 4 abutting each other, similar to observations of Ludwig et al. (2011).

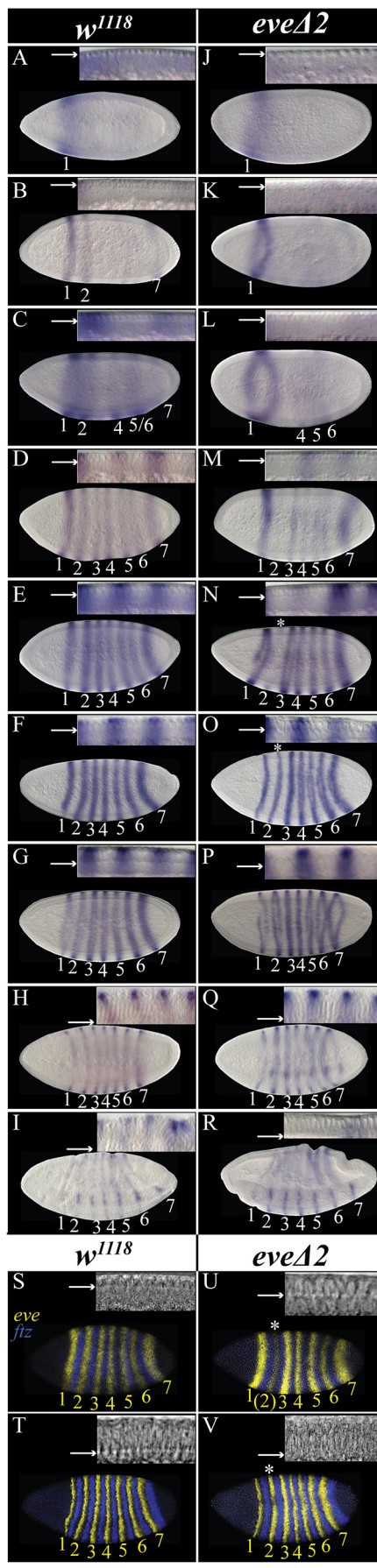

**Fig. 7. Early *eve* stripe 2 expression is initially perturbed in *eveΔ2* mutant embryos but 7-stripe expression recovers.** (A-R) Brightfield images of *eve* expression time course using digoxigenin-labeled probes in wild-type and *eveΔ2* embryos, as indicated. Embryos oriented anterior left, dorsal up except in H. Insets at upper-right corner of each panel show the progress of membrane invagination, with arrows indicating the leading edge of the membrane. Asterisks indicate reduced *eve* stripe 2 expression. (S-V) Fluorescence confocal images of *eve* and *ftz* expression detected by *in situ* HCR in wild-type controls (S,T) or *eveΔ2* embryos (U,V). *eve*, yellow; *ftz*, blue; Hoechst 34580, gray.

while stripe 2 and then stripe 7 arises (Fig. 7B). Next, stripes 1 and 2 intensify while weak stripe 4 expression appears and a broad band forms for stripe 5/6 (Fig. 7C). Next, stripe 3 appears, stripes 5 and 6 become defined stripes, and all other stripe intensities increase (Fig. 7D). At this point, the nuclei are fully elongated and cellularization begins. The expression levels in all seven stripes increase as the plasma membrane extends toward the basal plane of the nuclei (Fig. 7E) and all seven stripes reach peak expression as cellularization completes (Fig. 7F). Expression gradually decreases and following ventral furrow formation, the embryo enters the gastrulation phase, during which the *eve* stripes fade away (Fig. 7G-I).

The initiation of *eve* stripe formation in *eveΔ2* differs. Stripe 1 distinctly emerged first (Fig. 7J). Stripe 1 then became more defined, and a weak stripe 7 formed. This was followed by expression in the regions covering stripes 4, 5 and 6 (Fig. 7K). Next, stripes 3-6 became more defined and a faint stripe 2 arose on the ventral side (Fig. 7M). Subsequently, stripes 1 and 3-7 exhibited a continuous increase in intensity but stripe 2 was only marginally detectable (Fig. 7N, asterisk). This considerable reduction in stripe 2 expression in the mutant compared to the wild type was the primary difference between *eveΔ2* mutant and wild-type *eve* expression patterns. Similar to these results, an early, weak *eve* stripe 2 was also observed by Ludwig et al. (2005) using a rescue construct with eve2 deleted (see Discussion). As the plasma membrane extended toward the basal plane of the nuclei, all seven stripes intensified, and stripe 2 appeared as a full, clear stripe (Fig. 7O, asterisk), with all stripes persisting, then becoming thinner and fading during gastrulation (Fig. 7P-R). In summary, in *eveΔ2* mutants, the sequence of *eve* stripe formation was as follows: 1, 7, 4+5/6, 3 and 2, with stripe 2 exhibiting weak expression until near the end of cellularization, at which point it increased in intensity. Multiplexed fluorescence *in situ* HCR confirmed the low levels of *eve* stripe 2 during early stage 5 compared to wild type to be statistically significant (Fig. 7U, asterisk). This also confirmed recovery to wild-type levels during late stage 5 (Fig. 7V, asterisk; Fig. S10, Table S3). The width of *eve* stripe 2 was similar in wild type and *eveΔ2* mutants during both early-stage 5 and late-stage 5 embryos (Fig. S10). This suggests that, like *ftz*, the late-acting, 7-stripe CRE of *eve* can compensate for stripe-specific CRE mutations.

Despite this, the patterns of downstream target gene expression were not detectably altered in late *eveΔ2* mutants. In the absence of *eve*, *en* expression is lost in the odd parasegments and *slp1* expression is expanded (Frasch et al., 1988; Prazak et al., 2010; Schroeder et al., 2011). However, in *eveΔ2* mutants, expression of *en* (Fig. 6F,G) and *slp1* (Fig. S11) appeared wild type-like, although we cannot rule out small variations in stripe spacing or quantitative differences in expression levels of individual stripes. Thus, like *ftz*, the *eve* stripe-specific CRE contributes to early expression of that stripe but is not necessary for late expression of stripe 2 or for regulation of late Eve target genes.

### *eve* and *ftz* stripe 2 deletion mutants show reduced fitness

Although we were able to – without difficulty – maintain true-breeding, homozygous stocks of both *ftzΔ2/ftzΔ2 and eveΔ2/eveΔ2*, they may have reduced viability compared to wild type. To assess this, we collected eggs (100/replicate) from *w1118* controls, *eveΔ2/eveΔ2* homozygotes, or *ftzΔ2/ftzΔ2* homozygotes. From an average of four experiments, adult survival for *w1118* controls was 62.5%, while only 20% of *eveΔ2/eveΔ2* reached the adult stage. The *ftzΔ2/ftzΔ2* homozygotes fell somewhere in between with ~46% reaching adulthood, but this was still fewer than seen for controls (see raw data in Table S4). This effect was exacerbated at 28°C, with 44% of controls reaching adulthood but only~16% (*eveΔ2/eveΔ2*) and 10% (*ftzΔ2/ftzΔ2*) surviving. Decreased viability compared to controls was seen in both the number of hatched larvae and the number of pupae formed (Table S4). Thus, although they survive well as homozygotes in the lab, these mutants show reduced fitness compared to control flies.

We next assessed the strength of the *stripeΔ2* alleles by crossing the *ftzΔ2* or *eveΔ2* homozygotes to either weak or strong alleles of *ftz* or *eve*, respectively. Adult survival was scored (Tables S5, S6). *ftzΔ2* flies were crossed with either a weak *ftz* allele (*ftz5*, previously named *ftzf47ts*; BN2214) or to a *ftz* null allele (*ftz11*, previously named *ftzw20*; BN2224). From four independent crosses of virgin *ftzΔ2* homozygotes to male *ftz5/TM3,Sb,Ser*, expected to produce ~50% transheterozygotes, a total of 179 Sb (87 males, 92 females) and 231 non-Sb flies (103 males, 128 females) were obtained. Thus, *ftzΔ2/ftz5* survive at the Mendelian ratio expected (~50:50) if this transheterozygous combination is fully viable. In contrast, from four independent cross of virgin *ftzΔ2* homozygotes to male *ftz11/TM3, Sb*, again expected to produce ~50% transheterozygotes, the only surviving adults were Sb (171 males, 230 females; total 401 Sb flies). Thus, *ftzΔ2/ftz11* transheterozygotes do not survive to adulthood. In summary, flies carrying *ftzΔ2* in combination with a weak allele (*ftz5*) are viable but *ftzΔ2* in combination with a null allele is lethal. As *ftz5* itself is not homozygous viable, this suggests an allelic series ordered weakest to strongest: *ftzΔ2*, *ftz5*, *ftz11*.

Similarly, *eveΔ2* flies were crossed with either a weak *eve* allele (*eve1*, also known as *eve1D19*; BN5344) or to a strong *eve* allele that produces PR defects (*eve3*, also known as *eveR13*; BN299), or to a deficiency *eve* allele [*Df(2R)eve1.27*; BN1545]. Raw data for all crosses can be found in Table S6. From five independent crosses of virgin *eveΔ2* homozygotes to male *eve1/CyO*, expected to produce ~50% transheterozygotes, 846 Cy (383 males, 463 females) and 148 non-Cy flies (52 males, 96 females) were obtained. Thus, *eveΔ2/eve1* flies survived to adulthood but with less than the 50:50 ratio expected if the combination were fully viable. The reciprocal cross yielded even fewer *eveΔ2/eve1* surviving adults. For this, 870 Cy (426 male, 444 female) but only seven non-Cy adults (two male, five female) were obtained. Thus, a strong effect of parental allele origin was seen for transheterozygotes for these weak *eve* alleles. In contrast, from four independent crosses of virgin *eveΔ2* homozygotes to male *eve3/CyO*, again, expected to produce ~50% transheterozygotes, the only surviving adults were Cy (490 males, 595 females; total 1085 Cy flies). The reciprocal cross similarly yielded only Cy adults (581 males, 647 females; total 1228 Cy adults). Thus, *eveΔ2/eve3* transheterozygotes do not survive to adulthood. Finally, crosses were set up with the *Df(2R)eve1.27* allele. Irrespective of parental allele origin, only Cy flies were obtained, in keeping with the results for the *eve3* allele (in total, 1213 Cy flies). In summary, both *ftz* and *eve* stripe2 CRE deletion alleles behave as very weak hypomorphic alleles, which, when homozygous, result in animals that are viable and fertile but with reduced fitness.

To investigate the cause of lethality for *eve* transheterozygotes, large-scale crosses of virgin *eveΔ2* homozygotes to male *eve3/CyO* were set up and embryos were examined for cuticle defects and effects on target gene expression (Fig. 6E,H-J, Fig. S12). From this cross, 50% of the offspring should be *eveΔ2/eve3* transheterozygotes, while 50% should be *eveΔ2/CyO* and thus display wild-type segmentation. Clear defects were observed in larval cuticles for~half of the larvae (62 wild type-like; 53 non-wild type). Of the 53 showing defects, 50 showed anterior defects, with T1 decreased or missing (Fig. 6E, arrowhead). Of those showing anterior defects,~half (23) also showed A6 defects (Fig. 6E, asterisk). Three larvae showed a lawn phenotype, typical of severe *eve* mutants, possibly reflecting haploinsufficiency (see Fujioka et al., 2002; Hughes and Krause, 2001). Expression of *en* was also impacted in the primordia of this region, likely due to an overall decrease in the size of parasegment 3 documented by Ludwig et al. (2011) for similar eve2 mutants (see Discussion). Expression of *en* was altered in approximately half of the embryos (63 altered; 67 wild type-like), consistent with alterations in all *eveΔ2/eve3* transheterozygotes. *en* stripe 3 (Fig. 6H, arrow) or stripe 4 (Fig. 6I, arrow) were weak or undetectable, with other embryos exhibiting a decreased spacing between *en* stripes 3 and 4 (Fig. 6J, arrow), similar to that seen by Ludwig et al. (2011). These changes in the region of parasegment 3 were variable and, in some embryos, the spacing between other *en* stripes was also altered (Fig. S12).

### DISCUSSION

Here, we showed that the loss of stripe-specific CREs for the PRGs *ftz* and *eve* is compensated for by the presence of 7-stripe CREs. The contributions of 7-stripe and stripe-specific CREs differ in both time and space: they are regulated by different sets of transcription factors that are present in embryos at different time points and which, accordingly, activate or repress stripes at different stages of development. In addition, these TFs are distributed in different spatial patterns in the embryo, such that regulators of stripe-specific CREs only impact a subset of *ftz* or *eve* stripes while regulators of 7-stripe CREs are equally available to regulate all seven *ftz* or *eve* stripes. This compensation by different types of CREs is different from the redundancy provided by secondary enhancers (or shadow enhancers), which respond to the same regulatory cues as primary enhancers (Hong et al., 2008; Hobert, 2010; Perry et al., 2010; Barolo, 2012). Rather than responding to the same transcription factors, the 7-stripe and stripe-specific CREs 'get to the same finish line' – segmentation – but by different routes. For stripe-specific CREs, the exact combination and spatial domains of activators and repressors is unique for each PR stripe. In contrast, 7-stripe CREs are regulated by PR proteins themselves expressed in seven stripes. In most cases, these PR proteins act to refine the domains of other PRGs; for example, Hairy represses all *ftz* stripes after they are established (Howard and Ingham, 1986; Yu and Pick, 1995; Schroeder et al., 2011; Ingham and Gergen, 1988). In the case of direct autoregulation, best documented for *ftz*, the 7-stripe CRE (UPS) responds to activators: Ftz and Ftz-F1 activate all *ftz* stripes through the UPS. However, if 7-stripe CREs were functioning only to refine stripes, there would be no stripe to refine when a stripe-specific CRE is deleted, and that stripe would never develop. The fact that stripe 2 reaches wild-type levels for both *ftz* and *eve* after deletion of the respective stripe 2 CRE shows that 7-stripe CREs can establish stripes independently of the earlier action of stripe-specific CREs. This adds an order of complexity to the transcriptional regulation network that begs the question of how these two types of regulatory interactions arose and came to compensate for each other

during evolution [see discussion of these questions by Peel and Akam (2003) and Clark et al. (2019)].

## Stripe 2 CREs contribute to robustness

Given the role of *eve* stripe2 in promoting the development of a single stripe in response to non-periodic cues from maternal and gap proteins, it might have been expected that this CRE would be absolutely required for segment formation. The same holds true for *ftz* – although less well-studied, its stripe2 CRE likely responds to similar cues. Yet both deletion lines have been maintained in the lab for many generations as homozygotes, despite the fact that for both genes, no other stripe2 CRE has been identified. Interestingly, when subjected to 'genetic stress' both alleles proved to function as weak hypomorphs (Tables S5, S6), i.e. they do not produce sufficient product when present in single copy to complement null alleles. This is particularly interesting when we compare our results to those of previous researchers who used large rescue constructs to assess the roles of *eve* CREs (Fujioka et al., 1995, 1999, 2002; Ludwig et al., 2005, 2011). In that scenario, close to complete rescue of *eve* mutants was only achieved with two copies of the transgene. Flies carrying the rescue transgene with a stripe2 CRE deletion (Ludwig et al., 2011; p[*eve*$^{\Delta MSE}$]), removing the same region (minimal stripe element, see Fig. S9) removed in our study, were embryonic lethal, as were lines in which the endogenous eve2 was deleted but replaced with w+ (Ludwig et al., 2005, 2011; Fig. S5, *eve*$^{\Delta MSE}$). This apparent contrast with our results can be explained by relative allele strength. For the *eve* transgene, this quantitative effect is evidenced by the failure of single copy rescue. Interestingly, while *en* expression was wild type-like in *eveΔ2/eveΔ2* homozygotes, *en* patterns in *eveΔ2/eve³* transheterozygous embryos were similar to, although stronger than, that seen for both *eve*$^{\Delta MSE}$ (eve2 replaced with w+) and *eve* mutants carrying the p[*eve*$^{\Delta MSE}$] transgene (figure 3 in Ludwig et al., 2011 compared to our Fig. 6I,J and Fig. S12). We propose that the replacement of eve2 with w+ results in lower levels of *eve* product than our genomic deletion of eve2, which did not cause any further disruption to the locus. Similarly, deletion of a single *en* enhancer in the context of the genome was less impactful than its deletion from a transgene, likely because of the chromatin state of the native *en* genomic region (Cheng et al., 2023).

For *eveΔ2/eveΔ2* homozygotes, as well as *ftzΔ2/ftzΔ2* homozygotes, reduced viability compared to wild type was seen at both the standard fly-rearing temperature (25°C) and under heat stress (28°C; Table S4). Thus, for both genes, stripe 2 CREs serve to increase embryonic robustness but are not absolutely required for segment formation. It will be interesting to see whether the same findings hold true for other developmental genes with multiple enhancers that have distinct yet overlapping functions.

## MATERIALS AND METHODS
### Transgenic plasmids

Genomic fragments for reporter transgenes were amplified with NEB Q5 DNA polymerase (M0491S) following product specifications. Table S7 contains the sequences for all primers used in this work. All fragments were independently inserted in forward or reverse orientation into the XbaI site of vector *placZ-attB* using either Pyrite cloning (Fischer et al., 2018) or NEB HiFi Assembly (E2621S), to generate two reporter transgenes per fragment (forward and reverse). The sequence for each fragment and all *in situ* probe sequences are given in Table S8. Genomic DNA isolated from *nos-Cas9* flies was used as the DNA template for all PCR reactions unless specified otherwise. The *Drosophila* genome version dm6 was used as the reference genome (Dos Santos et al., 2015). Sequence fidelity of each construct was verified by comparing full sequence reads of each clone to an independently generated PCR product; sequencing by GENEWIZ. The ftz_6CRE

(ftz2+7) is between −6674 and −5435 and is 1240 bp. The ftz413 CRE (ftz2) is between −6418 and −6006 and is 413 bp. The ftz-7 CRE (ftz3+6) is between −8222 and −6669 and is 1554 bp. The ftz+3 CRE (ftz1+5) is between +2571 and +4333 and is 1763 bp. The 3′Element (3′E) CRE is between +1876 and +4008 and is 2133 bp. Transgenes were integrated into the phiC31 docking site at chromosome 2, 57F5, 2R:21645971 following embryonic microinjection into Bloomington *Drosophila* Stock Center (BDSC) line 9740 by Rainbow Transgenic Flies.

### Guide RNA and HDR template

Genomic targets for gRNA were selected using CHOPCHOP (Labun et al., 2019), the CRISPR Efficiency Predictor (Housden et al., 2015) and CRISPR Direct (Naito et al., 2015). A co-CRIPSR strategy to target *ebony* was utilized in all CRISPR experiments (Kane et al., 2017). Guide RNAs were inserted into the *pCFD5* vector, which processed all guides in a single transcript separated by an array of tRNA sequences (Port and Bullock, 2016).

Two gRNAs were selected to target the *ftz2* region. The left guide targeted the antisense strand from −6393 to −6412 to induce a double-stranded break 10 bp from the 5′ end. The right guide targeted from −6059 to −6040 to induce a double-stranded break 36 bp from the 3′ end. The *pCFD5* vector was used as a DNA template for PCR with the primers MF138 and MF139 in one reaction and MF140 and MF110 in another. The MF140+110 fragment was further amplified by MF140 and MF112. These two fragments were assembled into the *pCFD5* vector linearized by BbsI-HF using HiFi assembly to create *pCFD5-ftzΔ2*. Colony PCR was conducted with the primers CFDN-Seq and MF67. Sequencing of the isolated *pCFD5-ftz2* plasmid was conducted with the primer CFDN-Seq. The MIDI prep DNA used for microinjection was verified by digestion with XbaI and EcoRI-HF. The HDR template for the precise deletion of ftz2 (*pUC19-ftzΔ2*) was assembled from three PCR-amplified fragments. The left homology arm spans from −7792 to −6420 and was amplified by MF141 and MF40 using 0.625% DMSO and *pUC19-ΔUPS* as the DNA template to form a 1432 bp fragment. The right homology arm spans from −6006 to −3846, is 2153 bp, and was fused from two separate PCR reactions using *UPS⁰-F:lacZ* as the DNA template. The first reaction used the primers MF142 and MF143 with an adjusted extension temperature of 68°C and amplified a 572 bp fragment. The second reaction used the primers MF25 and MF144 with 0.625% DMSO and amplified a 1659 bp fragment. All three PCR fragments were assembled into *pUC19* linearized with KpnI with HiFi assembly to generate *pUC19-ftzΔ2*. Colony PCR was conducted with the primers Dm_UES_2F and MF143. *pUC19-ftzΔ2* was fully sequenced with the primers M13F, DmFtzUESSeq1F, MF142, MF25, DmUES3F and M13R. The MIDI prep DNA for microinjection was verified by digestion with EcoRI-HF.

Two gRNAs were selected to target the *ftz2+7* region. The left guide targeted the antisense strand from −6651 to −6670 to induce a double-stranded break 8 bp from the 5′ end. The right guide targeted the antisense strand from −5442 to −5461 to induce a double-stranded break 23 bp from the 3′ end. The *pCFD5* vector was used as a DNA template for PCR with the primers MF145 and MF146 in one reaction and MF147 and MF110 in another. The MF147+110 product was further amplified by MF147 and MF112. These two fragments were assembled into the *pCFD5* vector linearized with BbsI-HF using HiFi assembly to create *pCFD5-Δftz2+7*. Colony PCR was conducted with the primers CFDN-Seq and MF67. Sequencing of *pCFD5-Δftz2+7* was conducted with the primer CFDN-Seq. The MIDI prep DNA for microinjection was verified by digestion with XbaI and EcoRI-HF. The HDR template for the precise replacement of the full ftz2+7 fragment with only the ftz2 sequence (*pUC19-ftz2Δ2+7*) was assembled from three PCR-amplified fragments: a left homology arm, the ftz2 fragment and a right homology arm. The left homology arm spans from −7792 to −6676, is 1117 bp, and was amplified by the primers MF40 and MF94 with 0.625% DMSO using the *pUC19-ΔUPS* construct as a DNA template. The ftz413 fragment was amplified by the primers MF95 and MF96 with 0.625% DMSO using *UPS⁰-F:lacZ* as a DNA template. The right homology arm spans from −5435 to −3850, is 1586 bp, and was amplified by the primers MF97 and MF144 with 0.625% DMSO using *UPS⁰-F:lacZ* as a DNA template. All three fragments were assembled into the *pUC19* vector linearized by KpnI-HF using HiFi assembly. Colony PCR was conducted with the primers MF21 and MF96. The construct was fully

sequenced with the primers M13F, DmFtzUESSeq1F, MF24, MF97, DmUES3R and M13R. The MIDI prep DNA for microinjection was verified by digestion with EcoRI-HF.

To generate $ftz\Delta UPS^{7S}\Delta Z$ flies, the Z was deleted from flies homozygous for deletion of the $UPS^{7S}$. Homozygous $nosCas9;ftz\Delta UPS^{7S}$ flies were sent to Rainbow Transgenic Flies for embryonic injection with $pCFD4$-$U6$:$1$ $U6$:$3tandemgRNAs$ targeting the zebra element and the $pGEM$ HDR template targeting the zebra element (see Graham et al., 2021).

For $eve\Delta 2$, genomic targets for gRNA design were selected using CRISPR Direct (Naito et al., 2015). Two gRNAs were designed to target the $eve$ stripe 2 CRE. The left gRNA targeted positions −1535 to −1516, inducing a double-stranded break 22 bp from the 5′ end, while the right gRNA targeted positions −1105 to −1086, inducing a break 12 bp from the 3′ end. These guide RNAs were cloned into the $pCFD5$ vector, which expresses all guides as a single transcript separated by tRNA sequences (Port and Bullock, 2016). The vector was used as a template for PCR with two primer pairs: MF161/MF162 and MF163/MF110. A third fragment was generated using a single oligonucleotide, MF112. These three fragments were assembled into the $pCFD5$ vector, which was linearized with BbsI-HF, using HiFi assembly. Colony PCR and sequencing of $pCFD5$-$\Delta eve2$ were performed with primers CFDN-Seq and MF67. The homology-directed repair (HDR) template was assembled from two PCR-amplified fragments. The first fragment (1103 bp) was amplified using primers MF172/MF173 from −2629 to −1558, while the second fragment (1026 bp) was amplified with primers MF174/MF175 from −1073 to −80. Both fragments were inserted into the $pUC19$ vector, linearized with KpnI-HF, using HiFi assembly. Colony PCR was performed with primers MF164/MF165 to verify successful assembly. Embryos were injected with both the sgRNA array construct and the HDR template to generate $eve\Delta 2$ flies. To identify genomic deletions, gDNA was extracted from single flies, and 1 µl of the preparation was used for PCR with primer sets (MF164 and MF165) flanking the $eve$ stripe 2 CRE. The expected PCR product sizes for the wild-type and deletion alleles were 679 bp and 187 bp, respectively. PCR bands indicative of deletions were sequenced for confirmation.

### Drosophila genetics

*Drosophila* CRISPR/Cas9 protocols were the same as described by Graham et al. (2021). Fly stocks used were: $y[1]$ $M\{w[+mC]=nos$-$Cas9.P\}ZH$-$2A$ $w[*]$ (BDSC 54591), referred to as $nos$-$Cas9$ (RRID:BDSC_54591); and $P\{ry[+t7.2]=ftz$-$lacZ.ry[+]\}TM3$, $e$ $Sb[1]$ $ry[*]/Dr[Mio]$ (RRID: BDSC_3218), referred to as $Dr/TM3$, $Sb$ for balancing $ftz$ mutants, and $If/CyO$ for balancing the $eve$ mutant. $w^{1118}$ was used as wild type. To generate deletion lines, each injected $nos$-$Cas9$ adult was crossed to 3 $Dr/TM3$, $Sb$ or $If/CyO$ flies. Two or three additional backcrosses were carried out to eliminate the $nos$-$Cas9$ X chromosome.

Deletion of ftz2 to generate the $ftz\Delta 2$ allele was screened by PCR with the primers MF143 and Dm_UES_2F, which would amplify a 786 bp fragment from precise deletion events and 1198 bp from wild type (Fig. S2). Out of 32 fertile G0 lines, 12 produced progeny that tested positive for the precise deletion (37.5%). Precise replacement of the ftz2+7 region by ftz2 to generate $ftz2\Delta 2+7$ was screened by PCR with the primers MF21 and MF149, which would amplify 149 bp from HDR events and 405 bp from wild type (Fig. S3). Out of 24 fertile G0 lines, five produced positive progeny (20.8%).

The conditions for detecting deletion of the zebra element in the $ftz\Delta UPS$ background were as described for the original $ftz\Delta Z$ deletion (Graham et al., 2021). F1 progeny of injected G0 individuals were PCR-screened for the deletion. Primers were zebra1 and zebra2, which flank the deletion site (Fig. S4B, arrows indicate primer locations), amplifying an 888 bp fragment from individuals with successful deletion events and a 1485 bp fragment in the presence of the zebra element (Fig. S4C). DNA from individuals that tested positive was used as template in another PCR to be used for sequencing to confirm precise deletion of the element (Fig. S4D).

For morphological analysis of adult segmentation, adults were imaged under a Leica S6E dissection microscope.

### Cuticle preparations

Fly embryos were collected in cups with yeast paste for 24 h. Flies were flipped into fresh cups and the embryos aged for an additional 24 h. Larva

were poured into mesh baskets and rinsed several times to remove any residual yeast. The mesh baskets were treated with 50% bleach for 3 min to remove chorions, and the baskets were rinsed with water to remove bleach. The mesh was transferred to a 1.7 ml microfuge tube with 50% heptane and 50% methanol and shaken for 1 min. The heptane was removed and replaced with methanol, and the tubes were inverted to ensure mixing. Methanol was decanted and samples were rinsed with methanol three more times. Specimen were transferred to a slide and allowed to dry completely. Lactic acid was pipetted onto the sample and covered. The slides were placed in a 60°C oven for 2-24 h. Cuticles were imaged with darkfield microscopy using a Zeiss Microscope Axio Imager M1.

### Gene expression analysis

For colorimetric *in situ* hybridization, digoxigenin-labeled probes were used following standard protocols (Kosman et al., 2004) and imaged using a Zeiss Microscope Axio Imager M1 microscope. Probes were the same as used by Graham et al. (2021). Embryos homozygous for the lethal allele $ftz\Delta UPS^{7S}\Delta Z$ were identified with a multiplex *in situ* using a blend of antisense probes targeting both $ftz$ and $lacZ$ to determine which embryos had the third chromosomal balancer $TM3$, $Ser$, $hb>lacZ$. Embryos were evaluated for the pseudo-time series of stage 5 (Figs 3, 7) by zooming into the membrane on the dorsal side. Fluorescence *in situ* HCR was conducted, processed and quantified with ProStack (Surkova et al., 2019) and R as described previously (Graham et al., 2021; Fischer et al., 2024). In addition to the probes for $ftz$ and $eve$ for fluorescence *in situ* HCR described previously (Graham et al., 2021; Fischer et al., 2024), a $lacZ$ probe was utilized along with a hairpin amplifier conjugated to Alexa Fluor 647. All probes and fluorescent hairpin amplifiers for HCR were ordered from Molecular Instruments. Raw images were used for analysis and quantification.

Embryos used for HCR *in situ* hybridization were imaged on a Zeiss LSM 900 or 980 Airyscan confocal microscope with a 40× immersion oil objective. Six stacking optical sections, averaged eight times by line, were scanned through the lateral surface to a sufficient depth to detect all seven $ftz$ stripes (between 9 and 12 µm). Images were taken at a resolution of 1024×1024 pixels. Cellularization substage within stage 5 of embryogenesis was determined by the degree of cellular membrane deposition as viewed from the eyepiece with brightfield settings. Histograms for example images used in Figs 1, 3, 7 and Figs S1 and S3 were adjusted linearly in ImageJ for visualization. These linear adjustments were made uniformly across the entire image.

### Acknowledgements

We thank the lab of Dr Sougata Roy for the use of their confocal microscope and Amy Beaven for her assistance and guidance in the Imaging Core Facility. We acknowledge the Imaging Core Facility in the department of Cell Biology and Molecular Genetics at the University of Maryland, College Park for the use of the Zeiss LSM 980 Airyscan 2. Purchase of the Zeiss LSM 980 Airyscan 2 was supported by Award Number 1S10OD025223-01A1 from the National Institutes of Health. Thanks to Katie Reding and Ximena Gutierrez-Ramos for comments on the manuscript. We thank reviewer 1 for suggesting the genetic stress experiments, which gave such interesting results. Stocks obtained from the Bloomington *Drosophila* Stock Center (NIH P40OD018537) were used in this study. Some sections from the Results and Materials and Methods in this paper are reproduced from the PhD thesis of Matthew D. Fischer (Fischer, 2022).

### Competing interests

The authors declare no competing or financial interests.

### Author contributions

Conceptualization: M.D.F., L.P.; Data curation: M.D.F., K.A., M.L.; Funding acquisition: L.P.; Methodology: M.D.F., K.A., P.G., L.P., M.L.; Visualization: M.D.F.; Writing – original draft: M.D.F., K.A., L.P.; Writing – review & editing: M.D.F., K.A., P.G., L.P., M.L.

### Funding

This work was supported by the National Institutes of Health (R01GM113230 to L.P.). Open Access funding provided by the University of Maryland. Deposited in PMC for immediate release.

### Data and resource availability

All relevant data and details of resources can be found within the article and its supplementary information.

## The people behind the papers
This article has an associated 'The people behind the papers' interview with some of the authors.

## Peer review history
The peer review history is available online at https://journals.biologists.com/dev/lookup/doi/10.1242/dev.204872.reviewer-comments.pdf

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
