## [Peer Review File · Development (Cambridge, England)]

Compensatory action of different types of *cis*-regulatory elements buffers the striped expression of *Drosophila* pair-rule-genes

Matthew D. Fischer, Kristen Au, Minh Lê, Patricia Graham and Leslie Pick
DOI: 10.1242/dev.204872

Editor: James Briscoe

Review timeline

Original submission:	24 April 2025
Editorial decision:	16 June 2025
First revision received:	21 October 2025
Editorial decision:	8 November 2025
Second revision received:	12 November 2025
Accepted:	12 November 2025

Original submission

First decision letter

MS ID#: dev.204872

MS Title: Compensatory action of divergent *cis*-regulatory elements buffers the striped expression of *Drosophila* pair-rule genes

Authors: Matthew D. Fischer; Kristen Au; Patricia Graham; Leslie Pick
Article Type: Research Article

Dear Dr Pick,

I have now received all the referees' reports on the above manuscript, and have reached a decision. The referees' comments are appended below, or you can access them online: please go to:

As you will see, the referees express considerable interest in your work, but have some significant criticisms and recommend a substantial revision of your manuscript before we can consider publication. Please address the apparent discrepancy with previous studies by Ludwig et al. (2005, 2011) and discuss how your findings relate to this earlier work. Testing whether stripe 2 enhancers are required for developmental canalization by subjecting flies to environmental stress (higher temperatures) or genetic stress (crossing to other alleles) would be a valuable extension to the paper. I also highlight the comments requesting more precise developmental staging information, particularly clarifying when during stage 5 each line activates stripe 2 expression. Finally, I agree with the referee that the quantitative claims about expression levels, timing, and spatial patterns would benefit from more rigorous methodology. I realise that implementing single-molecule FISH etc is beyond the scope of the current study, but acknowledge these limitations would be important. In addition, each of the referees has several constructive suggestions to help improve the clarity of your study.

If you are able to revise the manuscript along the lines suggested, which may involve further experiments, I will be happy to receive a revised version of the manuscript. Your revised paper will be re-reviewed by one or more of the original referees, and acceptance of your manuscript will depend on your addressing satisfactorily the reviewers' major concerns. Please also note that

Development will normally permit only one round of major revision. If it would be helpful, you are welcome to contact us to discuss your revision in greater detail. Please send us a point-by-point response indicating your plans for addressing the referees' comments, and we will look over this and provide further guidance.

Please attend to all of the reviewers' comments and ensure that you clearly highlight all changes made in the revised manuscript. Please avoid using 'Tracked changes' in Word files as these are lost in PDF conversion. I should be grateful if you would also provide a point-by-point response detailing how you have dealt with the points raised by the reviewers in the 'Response to Reviewers' box. If you do not agree with any of their criticisms or suggestions please explain clearly why this is so.

Reviewer 1

SUMMARY OF THE ADVANCE MADE IN THIS PAPER AND ITS POTENTIAL SIGNIFICANCE TO THE FIELD

In this nice study the authors rigorously dissect the role that individual pair rule stripe enhancers play in specifying the body plan of the *Drosophila* embryo. Focusing first on *ftz* they identify its stripe 2 enhancer, and then carefully test its function by deleting it in the context of the endogenous locus using CRISPR/Cas9. They show that even though this alters the early expression of *ftz*, the expression of key *ftz* target genes remain essentially unchanged. In addition, under lab conditions the flies are viable and show no significant defects in segmentation. From previous work it was known that two other 7 stripe enhancers also drive expression in the stripe 2 domain (UPS & Z) and so using a similar approach the authors simultaneously deleted these and assessed the impact this has on patterning. They showed that while this compromised viability, and lead to defects in segments associated with other stripes, the stripe 2 enhancer on its own is sufficient to specify correct target gene expression around it and direct proper segmental fate. To test the generalizability of their findings, they turned to *eve* and deleted the famous *eve* stripe 2 enhancer from the endogenous locus. Surprisingly, removing this enhancer has no discernable impact on the expression of *eve* targets, segmentation, or viability. This is even though the deletion causes a clear loss of *eve* expression in the stripe 2 domain until quite late in *nc14*.

These results are significant, novel and likely to be of broad interest to the developmental biology community, given that they challenge the conventional wisdom of how one of the model patterning systems in development works. The paper is well written, and the data reported clearly justify the conclusions drawn.

SUGGESTIONS TO AUTHORS

The only major issue I have is that previous studies where the *eve* stripe 2 enhancer was deleted in a rescue context are not cited or discussed (Ludwig M. Z., et. al., 2005 PLoS Biology & 2011 PLoS Genetics). In contrast to this work, these past studies found that the enhancer was essential for viability in the context of a transgene capable of rescuing *eve* null mutant flies to fertile adulthood. However, this apparent discrepancy can be reconciled by the fact that their unmodified transgene behaved like a hypomorph, showing reduced viability, especially as a hemizygote.

While I don't think this is necessary for publication, I do believe the authors have an opportunity to significantly extend the scope of the study with some straightforward additional experiments. In my opinion these have the potential to elevate this study from being a nice paper to a textbook example.

The authors already show that the removal of the stripe 2 enhancers of both *ftz* and *eve* are phenotypically buffered for flies kept under ideal lab conditions. Yet both enhancers are deeply conserved, which begs the question are they playing a functional role or are they merely vestigial? In the past several decades multiple examples in fly (*sna*, *svb*, etc.) and mouse (*gli3*, *shox2*, etc.) development have come to light where seemingly redundant enhancers (with expression patterns that partially overlap in space and/or time) have been shown to be required for canalization. In the absence of environmental or genetic stresses individuals missing single enhancers typically showed no phenotypes, similarly to here. Yet, as soon as they are subjected to stress individuals missing enhancers show severe phenotypes, while those with both are typically fine.

To test if either ftz stripe 2 or eve stripe 2 are required for canalization the authors would simply need to raise their flies at higher temperatures and see if they start to see denticle belt defects. A genetic stress could be applied by crossing the enhancer mutants to ftz or eve nulls and observing if hemizygotes lacking the stripe 2 enhancers start to show segmentation defects. Definitively showing one of the most heavily studied enhancers in development is required for canalization, and that this is also true for other pair rule genes, would significantly extend the scope of the study.

Minor Points:

- 1) About midway down page 3 there is a sentence "... hour period after egg laying", this should be "... hour period after fertilization", since fly embryos can start developing before the eggs is deposited.
- 2) At the top of page 13 the paragraph ends with "for regulation of Eve target genes". I think this should be more accurately stated as "late Eve target genes". Eve has other targets (e.g., ftz, odd & run) that it regulates before en and slp1 (which come on quite late) but how their expression might change is still unclear.
- 3) The first sentence in the Drosophila genetics section on page 19 could do with some rewording for clarity.
- 4) On page 20 it is stated that "Cuticle preparations used standard methods." This is too vague, either a clearer description is required, or an established protocol should be referenced.
- 5) Midway on page 20 "conjugated to Alex Fluor 647", should read "conjugated to Alexa Fluor 647"

Reviewer 2

SUMMARY OF THE ADVANCE MADE IN THIS PAPER AND ITS POTENTIAL SIGNIFICANCE TO THE FIELD

Pair-rule gene expression is a key model for transcriptional regulation, developmental patterning, and the evolution of development; in particular, the eve stripe 2 enhancer and the ftz zebra element are textbook examples that most developmental biologists will have passing familiarity with. However, since eve and ftz both have both stripe-specific and zebra-type enhancers, the regulation of these genes is more complicated than the textbook model, as pointed out by the lead author of this study 30 ago (Yu and Pick 1995).

In recent years it has become possible to study the function of specific enhancers by using CRISPR/Cas9 plus homology-directed repair to make precise genomic deletions, an advance over earlier approaches using transgenic reporter genes and (partial) rescue constructs. The Pick lab has previously investigated the effects of deleting the ftz zebra element (Graham et al 2021) and the ftz upstream element (Fischer et al 2024) individually. In this manuscript they delete the stripe 2 enhancer of ftz, the stripe 2 enhancer of eve, and make a double deletion of both ftz 7-stripe elements (zebra and UPS). The double deletion allows them to look at the function of the ftz stripe-specific enhancers in the absence of the zebra enhancers, while the stripe 2 enhancer deletions allow them to ask the reciprocal question for eve and ftz stripe 2.

These new fly lines enable a nice clean test of stripe-specific enhancer function vs zebra enhancer function. The stripe 2 enhancer deletions result in loss of early stripe 2 expression for both eve and ftz, while the 7-stripe double deletion results in loss of late ftz expression in all stripes. Thus, as expected from reporter studies, the enhancers drive expression in different temporal ranges, which overlap at late stage 5. Losing the ftz 7-stripe elements has stripe-specific effects on gene expression and downstream segment patterning which are anti-correlated with contributions from the stripe-specific elements (e.g. stripe 4 which has no stripe-specific element is worst affected, but several of the other stripes/segments are fine); this is also as expected. Loss of the ftz stripe 2 enhancer produces no apparent effects on late ftz expression or downstream segmentation; this is not necessarily expected (why then is the ftz stripe 2 enhancer present - perhaps it provides

patterning robustness under less ideal environmental conditions?) but it is consistent with the known roles of Ftz in the segmentation network, given that Ftz is not important for early patterning of other pair-rule genes, and its key targets (e.g. various segment-polarity genes) are not expressed until late stage 5.

The headline result of the study, however, is that the eve stripe 2 enhancer is dispensable for viability and normal segmentation (though does seem to contribute to patterning robustness, given the low frequency thoracic defects observed in the deletion mutants). This is really surprising! Not only because eve stripe 2 is one of the most famous and best studied developmental enhancers, but because the early eve stripes have previously been shown to play important roles in pair-rule gene cross-regulation. Indeed these new results for eve stripe 2 contrast with earlier studies which found that the early expression from the eve stripe 4+6 element is required for normal segmentation (Fujioka et al 1995, Fujioka et al 2002). This work therefore suggests that the importance of the early Eve stripes differs from stripe to stripe, perhaps correlating with how the stripes of other primary pair-rules are regulated in the vicinity (more by stripe-specific elements or more by cross-regulation through 7-stripe elements).

A limitation of the study is that it only looks at the segmentation markers *en* and *slp*, and only at extended germband stage rather than late stage 5/stage 6 when they are first expressed; the immediate impact of the enhancer deletions on downstream segmentation gene expression is therefore unclear. However, the key finding is that both stripe 2 deletion lines give rise to viable adults with normal segments, and this is clearly justified by the data shown.

SUGGESTIONS TO AUTHORS

My suggestions involve changes to the text and/or presentation of the figures.

Major comments:

1. Title/summary/abstract/discussion: the authors have chosen to highlight the functional redundancy of the *ftz* stripe 2 and 7-stripe elements and their expression overlap at late stage 5 as the take-away message of the paper. However, this finding for *ftz* stripe 2 does not generalise (e.g. it is not the case for *ftz* stripe 4 or 5) and in my opinion it also buries the lead result of the paper, which is that the eve and *ftz* stripe 2 enhancer deletions do perturb the early stripes, but with little apparent effect on subsequent development. The authors might consider revising/reframing the relevant parts of the text to make both sets of conclusions clear.
2. Embryo staging within stage 5: the Methods states (L601) that "Cellularization substage within stage 5 of embryogenesis was determined by the degree of cellular membrane deposition as viewed from the eyepiece with brightfield settings". It would be useful to have this staging information provided alongside the embryos shown in Figure 3 and Figure 7, similar to e.g. Figure 2 in Schroeder et al 2011. This information is included in the main text for the eve stripe 2 deletions (L340-352), but not for the *ftz* enhancer deletions. As such, it is not clear the extent to which the rows in Figure 3 represent equivalent developmental timepoints, nor how to relate Figure 3 to Figure 7. This is particularly important because Schroeder et al 2011 show the eve late element driving expression in stripe 2 much later than the *ftz* LacC element (30-40' vs 5-17' into stage 5). However, both Fig 3 and Fig 7 show the stripe 2 deletion lines activating stripe 2 in the 5th row out of 9, giving the impression they turn on at the same time. When exactly during stage 5 does each line activate stripe 2, and is this consistent with the reporter gene expression described in Schroeder et al 2011?
3. Existing literature: a very pertinent previous study should be referenced and discussed: Fujioka et al (2002) *Development* 129, 4411-4421 - see in particular Figure 3. Fujioka et al separately deleted the eve 4+6 enhancer and the eve late element from an eve rescue transgene in an eve null background, allowing them to assess how these stripe-specific and 7-stripe elements contribute to patterning. (See also the earlier experiments investigating early vs late Eve function in Fujioka et al 1995.) For the eve 4+6 deletion, the thin late stripes still appeared in stripes 4 and 6 despite the loss of the early stripes, similar to the stripe 2 deletion described here. However, the late stripes only partially rescued segmentation and the resulting flies, although viable, had two fewer abdominal segments. (The earlier work in Fujioka 1995 suggests that the segmentation defects are

mediated by misexpression of genes that would normally be repressed by Eve, such as runt, prd and slp.) The different result found here for the eve stripe 2 deletion is very interesting; the stripe-specific outcomes of these SSE deletions should be highlighted and discussed. For the late element deletion, late eve expression was lost as expected. Some of the lines rescued ~10% of individuals to adulthood, but there was a derepression of slp into the cells that would normally express late eve, showing the importance of this element for stabilising the parasegment boundaries.

Additional minor comments:

Title: consider word choice. "divergent" has evolutionary connotations and might give the misleading impression that the stripe-specific and 7-stripe enhancers are homologous to each other. Similarly, "compensatory" might suggest that one enhancer upregulates or downregulates expression depending on the presence of the other enhancer.

L46: "the classic eve stripe 2 CRE is not necessary for the expression of eve stripe 2" - this is only true for the late expression, and even this seems to be thinner than the wild-type expression (see comment on L362). "...or for Eve's regulation of downstream target genes": we do not know this to be the case for the early regulation for en and slp, nor for earlier expressed target genes such as odd and prd.

L64: "In their absence, segments do not form." - for most pair-rule gene mutants, segments still form, just only half of them.

L65: "arising and then rapidly fading during an ~2 hour period after egg laying." - 2 hrs AEL would be the beginning of stage 5, before stripe formation.

L96: "A Late-acting (L) autoregulatory 7-stripe CRE was also identified for eve" - subsequent studies argue that this enhancer is only indirectly autoregulatory, i.e. repressed by genes that are themselves repressed by Eve: see e.g. Manoukian and Krause, Genes and Development (1992) p.1748, section "Regulation of the en and eve genes".

L113: "odd-skipped (odd) and runt (run) contain stripe-specific elements for all stripes" - the paper cited (Schroeder et al 2011) actually says that odd lacks SSEs for stripes 2, 4, and 7.

L114: "sloppy-paired (slp) and paired (prd) are regulated in a stripe-specific fashion for stripe 1, but only 7-stripe CREs have been identified for the remaining stripes" - prd has a SSE covering both stripes 1 and 2 (Ochoa-Espinosa et al 2005) and also has regulatory sequences specifically associated with the posterior part of stripe 2 (Gutjahr et al 1993).

L130: "These studies" - unclear which studies are being referred to as only Graham et al 2021 is cited in the previous sentence.

L153 (and other instances): "ftz-6" - Schroeder et al 2011 used the slightly different name ftz₍₋₆₎.

L173: "ftz-6R$\gt;lacZ$ was expressed in a strong stripe overlapping ftz stripe 2 and a very weak (near background levels) stripe overlapping ftz stripe 7" - for ftz₍₋₆₎, Schroeder et al 2011 show stripe 7 appearing considerably later (phase 3) than stripe 2 (phase 1), but the late stripe 7 doesn't look particularly weak. Was stripe 7 very weak even in older embryos?

L176: "The weaker expression of stripe 2 for the ftz2+7 transgene compared to the ftz2 transgene" - it's unclear from the methods how this quantitative assessment was made. E.g. were staining reactions run simultaneously for both reporter lines and stopped after a standard time?

L186: "A second deletion was generated to test whether the regions flanking ftz2 that are present in ftz2+7..." - without studying Fig S3 it's unclear from this description that the deletion removes the just the flanking regions while retaining the ftz2 enhancer, so this could be stated more explicitly in the main text.

L199: "we asked if: 1) the combined input of these 7-stripe CREs is necessary for segmentation and 2) stripe-specific CREs are sufficient to establish segmentation without any 7-stripe CRE contribution" - the distinction between these questions is not clear to me.

L214: "for most ftz stripes, stripe-specific CREs can provide sufficient levels of ftz for wildtype-like function." - and presumably a sufficient spatial refinement of the stripe as well.

L231: "peak expression is reached for all seven ftz stripes" - was this measured? Include details in methods.

L269: "At the next time point" - see major comment 2 re embryo staging.

L279: missing word: "during the blastoderm [stage]"

L287: "reduced but still detectable in either ftz Δ Z or ftz Δ UPS7S homozygotes" - add references to this previous work.

L302: "It appears that the proper development of *Drosophila* is refractory to losing a stripe-specific CRE." - unclear as written whether this is meant to be a general statement or refers only to stripe 2. See also major comment 3.

L306-328: It is not clear to me from the text how often defects were found in regions corresponding to ftz stripes 2, 3, and 6. E.g. line 315: "Though the exact expression pattern varied between embryos" - does this include defects related to stripes 2/3/6 or only 4/5/7? If the former, the conclusion that "ftz stripe-specific CREs alone are sufficient for consistently directing segmental fate for T2, A1, and A7" needs to be qualified, due to the loss of patterning robustness observed. There do seem to be slp and/or en abnormalities around parasegment boundaries 4/6/12 in Fig 5D/E/I/J. Note also that changes to the duration or spatial distribution of ftz expression may be contributing to the segmentation phenotypes, in addition to the expression level changes discussed.

L310: "this corresponds to decreases in levels of ftz stripes 4 and 5" - it would be helpful for readers to have a reference diagram relating the locations of the ftz stripes to those of the en/slp stripes and the eventual segments of the larval cuticle.

L312: "slight expansion of individual stripes" - this points to a more general role for the ftz 7-stripe elements in maintaining the stability of the parasegment boundaries, as found for the eve late element (see major comment 3 above).

L326: "Thus, it appears that ftz stripe-specific CREs alone are sufficient for consistently directing segmental fate for T2, A1, and A7. However, A3 fails to form (there is no stripe 4 CRE), and A4 is consistently fused to A5." - again, it would be useful for the reader to have a reference diagram relating the Ftz stripes to the eventual segmental boundaries.

L338: "Minor defects in the thorax were observed in some cases" - this suggests that the early stripe might increase the robustness of patterning, even if most embryos can do without it.

L341: "Like ftz, establishment of the eve stripes occurs in distinct phases" - with the caveat that Surkova et al 2008 found that (protein) expression patterns of both genes varied somewhat within timeclasses, and the order of stripe emergence also showed some variation.

L360: "early stripe 7 seemed to be weaker than wildtype" - longer versions of the eve stripe 2 enhancer drive expression in stripe 7 as well [see e.g. Staller et al, PNAS (2015) and references therein], consistent with this observation.

L362: "and stripe 2 appeared as a full, clear stripe (asterisk, Fig. 70)" - it's difficult to judge conclusively from the colorimetric in situ, but this stripe appears to be narrower than the wildtype stripe 2 in 7F, as would be expected from the known expression pattern of the eve late element relative to the SSEs.

L368: "Despite this, downstream target gene expression was not disrupted in *eveΔ2* mutants." - see comment above on L46.

L371: "in *eveΔ2* mutants, expression of *slp1* and *en* were both indistinguishable from wildtype" - it's hard to judge from the colorimetric in situ, but it looks possible that the spacing between stripes 3 and 4 is narrower than in wild-type.

L373: "or for regulation of Eve target genes." - see comment on L46.

L384: "which respond to the same regulatory cues as primary enhancers" - I don't think this is generally how the terminology is understood - see e.g. Staller et al, PNAS (2015) for a counterexample. The key criterion would be overlapping expression, not similar regulation.

L386: "the 7-stripe and stripe-specific CREs 'get to the same finish line but by different routes.'" - however their expression patterns only overlap transiently in time, and for the *eve* late element the spatial overlap with the SSEs is only partial.

L394: "Hairy represses all *ftz* stripes after they are established" - *ftz* derepression begins quite early in hairy mutants; I'm not sure all the *ftz* stripes are established beforehand.

L395: "In the case of autoregulation, 7-stripe CREs respond to activators" - see comment above for L96.

L397: "The existence of these two different routes to stripe expression was thought to fall neatly into a two-tier system, with stripe-specific CREs establishing stripes and 7-stripe CREs refining them." - add references to illustrate this.

L397-403: "...The fact that stripe 2 reaches wildtype levels for both *ftz* and *eve* after deletion of the respective stripe 2 CRE shows that 7-stripe CREs can establish stripes independently of the earlier action of stripe-specific CREs." - given that secondary pair-rule genes lack SSEs for most stripes and primary pair-rule genes can lack SSEs for specific stripes (e.g. *ftz* 4, *odd* 2, 4, 7), this was already known.

L404: "begs the question of how these two types of regulatory interactions arose and came to compensate for each other during evolution." - this has been discussed in the context of the evolution of long-germ segmentation from an ancestral short-germ state. See for example Fig 2 in Peel and Akam 2003 or Fig 5 in Clark, Peel and Akam 2019.

L414: "downstream genes are not activated until a certain time" - the role of *Opa* in activating late segmentation gene expression is relevant here (see e.g. Clark and Akam 2016, Soluri et al 2020, Koromila et al 2020).

L416: "peak-level expression in stripes at the end of cellularization/start of gastrulation (end stage 5/beginning stage 6) is crucial. Their expression earlier (as shown by stripe-specific CRE deletions) or later (as shown by the UPS deletion) is of no consequence" - this might be true for *ftz* (though see comment for L312 above) but it is not generally true for other primary pair-rule genes. *eve* and hairy mutants both show significantly perturbed downstream gene expression at mid stage 5, for example. See also major comment 3 above.

L437-440: The meaning of "provides enough regulatory information" is ambiguous; the intended meaning seems to be "drives enough expression".

L457: "it has an autoregulatory role" - see comment above for L96.

L459: "although this remains to be tested, it is likely that the *eve* Late CRE is as dispensable as the *ftz* UPS" - see major comment 3.

L600: "Images were taken at a resolution of 1024 px²." - 1024 x 1024 px.

L861: typo: "partial f A4/A5 usion"

L884: "Brightfield images of ftz expression time series throughout stage 6 (gastrulation)" - the embryos in rows 2-4 seem to be at stage 7.

L901: "(D) Wildtype cuticles" - typo, should be (C).

L904: "Downstream gene expression is not perturbed in eve Δ 2 mutants" - see comment for L371.

L917: "(K) A weak stripe 7 start to appear along." - part of sentence missing?

Figure 1

- It could be clearer on the diagram that ftz2 is a renamed 413 and ftz2+7 is a renamed ftz-6.
- Consider colour-blind friendly colour combinations for the merges in D and G.
- Labelling the eve and ftz channels on the figure (not just in the legend) would be easier for readers.

Figure 3

- See major comment 2.
- Adding annotations to number emerging stripes might be helpful.
- There is some variation in embryo orientation which could be noted in the legend.
- The ventrolateral embryo in CG should be flipped vertically.

Figure 7

- See major comment 2.
- As with Fig 3, there is some variation in embryo orientation which could be noted in the legend.
- Some embryos should be vertically flipped, e.g. M.
- Green / magenta would be more colourblind-friendly than green / red.
- eve and ftz channels could be labelled on the figure.

Supplementary figures:

- Add scale annotations for the ladders (S2-4, S6)
- Fig S3, panel D: it is hard to relate the description in the legend ["405 bp fragment for negative events (lanes 6-7)"] to the figure because the obvious band in lanes 6-7 is for a shorter fragment; add additional annotations or explanation as appropriate.
- Fig S4: state what ladder was used in legend. Add a sentence to the legend to explain why both bands are present (i.e. flies are heterozygous over balancer).

Reviewer: Erik Clark

Reviewer 3

SUMMARY OF THE ADVANCE MADE IN THIS PAPER AND ITS POTENTIAL SIGNIFICANCE TO THE FIELD

Compensatory action of divergent cis-regulatory elements buffers the striped expression of *Drosophila* pair-rule genes

This manuscript investigates the regulation of ftz expression in the fly embryo and the functional consequence of enhancer deletions. The authors compare two types of enhancer deletions: stripe specific CRE mutants (delta stripe 2 or delta stripe 2+7) with a larger deletion of two entire regulatory regions UPS and Z, known to operate as two separate 7-stripe CREs. Through mostly fixed approaches, they investigate the timing, location and levels of ftz expression (as well as Ftz targets) in these various enhancer deletions. They also perform a phenotypic analysis to examine the functional consequences of enhancer deletions.

Finally, the authors compare ftz regulation to that of eve. Indeed, eve is supposed to rely on stripe specific regulations while ftz was thought to be regulated by a two 7-stripe CREs. Surprisingly, they found that deletion of eve stripe 2 enhancer has no functional impact on embryogenesis. As observed for ftz, the deletion of eve stripe 2 enhancer initially perturbs eve expression in this domain, but there are compensatory effects such that by the end of nc14, eve expression appears normal.

The main conclusion of this work is that 7-stripe CREs can compensate for the loss of a stripe a specific enhancer, even if these 2 types of enhancers are regulated by different input. They propose a model in which a periodic pattern can be established by two distinct manners (1 stripe at a time that is unprecise then sharpened or 7 stripes regulated by TF which are themselves expressed in a periodic manner). These two types of regulations can compensate for each other, thus providing redundancy to the system.

Overall, the manuscript investigates an original question regarding spatio-temporal regulation of transcription by enhancers and the regulation of *ftz* stripes along the fly embryos A/P axis is the best system to address this question. I appreciate the systematic attempt of the authors to link gene expression levels to functional output with phenotypic analyses. My major comment is that the questions of transcription dynamics (levels, timing and spatial coordinates) require a more quantitative approach than colorimetric *in situ* hybridization. It seems difficult to apprehend the questions asked without subcellular resolution of mRNA production and without a dynamic approach. Moreover, I found the manuscript (and figures) difficult to read.

SUGGESTIONS TO AUTHORS

Major Comments

Figure 1:

I assume colorimetric ISH are with a *lacZ* probe. This should be mentioned.

Why is the *ftz2* Reverse transgene more efficient than the forward in driving expression in stripe 2? (Colorimetric signal in B versus C)

Ftz6 was renamed *Ftz2+7*, but the expression in stripe 7 is very weak: it can only be observed with the colorimetric ISH, and only in the reverse position. Can the author comment on this?

Figure 3 reconstitutes the timing of appearance of *ftz* expression in each stripe, in various enhancer deletion contexts. Nowadays, this temporal question is generally examined with live imaging, with MS2 reporters, a technique widely employed to study gene expression dynamics in the early fly embryo.

Indeed, colorimetric ISH is not quantitative, yet the description and interpretation of the entire figure 3 relies on expression levels. If live imaging is challenging to implement for this lab, a FISH (ideally smFISH now inexpensive with techniques as smiFISH, for example Calvo et al, <https://doi.org/10.1038/s42003-021-01803-0>) could be used, with *ftz* as well as an internal marker to better label the timing of *nc14* progression.

I assume that panels of Figure 3 are ordered according to *nc14* progression (for example AA being younger than AD, again this nomenclature is confusing and not explained). However, without a labeling of the membrane invagination or an endogenous transcript, this temporal ordering remains challenging.

-From Figure 3, the authors conclude that deleting *ftz2* enhancer or *ftz 2+7* enhancer affects the early expression of stripe2 but this effect is somehow compensated by the end of *nc14* (stage5). However, such a conclusion cannot be drawn without a rigorous quantification of stripe2 pattern. How precise are stripe 2 anterior and posterior borders? How comparable are TS intensities in the absence of *ftz2* or *ftz2+7* enhancers? Colorimetric ISH is not quantitative enough to conclude on transcription levels, timing nor spatial precision.

-Figure 4 is the mirror Figure of 3 but examined at later stages (stage6). It shows a clear absence of *ftz* expression upon deletion of the UPS and Z control regions.

-Figure 5: same comments as for Figure 3 and 4. The levels of expression could be more rigorously assessed with a more quantitative approach (smFISH or live imaging).

-Figure 6: as for Figure 3, side by side comparison can be achieved only if shown embryos are at the same stage. Indeed, *eve* expression is particularly dynamic in *nc14* (Bothma 2014). If embryos are of comparable stages, *eve* expression in stripe 2 appears perturbed not only at early stages but also in cellularized embryos: panel O compared to F. In fact, stripe 1 also appears thinner in O, compared to F. Panel P should be replaced.

Minor Comments

Figures are difficult to read.

Some suggestions: split the colorimetric ISH and the fluo ISH into distinct panels with a clear white border. The genotype/ name of transgene could be indicated above. The nomenclature *ftz-2F-hsp70:lacZ* is not easy to read.

Panel A: maybe no need to indicate the sizes?

The manuscript is also not easy to read. I would avoid using too many acronyms in the abstract (PRG, CRE). The manuscript also contains many acronyms and jargon (UPS, Z, primary or secondary PRG, PR-stripe) and should be reformulated to be understood by a broad readership. It's unclear to me who are the primary and secondary PRG.

To understand which cis-regulatory information is kept upon deletion of the UPS and Z elements, these regions should be indicated in a summary figure of the *ftz* locus.

First revision

Author response to reviewers' comments

We thank the reviewers for their interest in our work and for very helpful suggestions and critiques. Our responses are in blue. Please note that line numbers referred to in our response match the Highlighted text version to allow reviewers to see the changes made in response to their suggestions, and will be slightly different from the line numbers in the cleaned up version of the text. In that versions, deleted text is highlighted with magenta and added text is highlighted with green.

In addition to the specific responses below, please note that because we added the new experiments suggested by Reviewer 1, we had to remove previous text, mostly from the Discussion and Figure Legends, to stay within the 7000 word limit required by Development.

We have made changes to figures throughout the manuscript:

Figure 1:

- Simplified the naming in the schematic in panel A
- Removed “hsp70” from the transgenic schematics in panels B-G
- Changed the FISH images from *eve*-red *ftz*-green Hoechst-blue to the more colorblind-friendly *eve*-yellow, *ftz*-blue, Hoechst-gray. This change has been made to every FISH image throughout the manuscript to maintain consistency and colorblind-friendly format.
- *eve* and *ftz* are now written color-coded near the FISH images.
- Edit legend to indicate that probes targeting *lacZ* were used in the colorimetric in situ.

Figure 2:

- A different image has been used for wild type adult, as we realized it was from a different data set. The phenotype is identical and does not change any of our descriptions or conclusions.
- For this figure and throughout the manuscript, we renamed *ftz2Δ2+7* to *ftz2ΔFlanking* to be more intuitive as to what this genotype is (see Fig. S5 for schematic)

Figure 3:

- Numbers have been added to identify the *ftz* stripes in all panels.

- Insets of the plasma membrane on the dorsal side of the embryo have been added to each panel. An arrow indicates the degree of membrane deposition to determine the placement in the pseudo-time series.
- Removed the original panels AC, BB, CC, DC, as we determined they were superfluous
- Re-oriented embryos to be dorsal-top and ventral bot (ex: the original panel CG, now CF). We have also edited the legend to indicate
- Added HCR FISH images of each genotype as panels AI, BI, CI, and DI. These represent late stage 5 embryos.

Figure 4:

- No change to figure; legend edited to specify which rows are stage 6 and which are stage 7.

Figure 5:

- No change

Figure 6:

- A different image has been used for wild type adult, as we realized it was from a different data set. The phenotype is identical and does not change any of our descriptions or conclusions
- New *engrailed* chromogenic in situ stains from *eveΔ2/eve3* transheterozygotes have been put into panels H, I, and J
- We have moved the original *slp1* staining from WT and *eveΔ2* to the supplement (Fig. S11)

Figure 7:

- Like figure 3, we have added numbers to indicate the *eve* stripes
- Like figure 3, we have added to each panel insets of the plasma membrane on the dorsal side of the embryo. An arrow indicates the degree of membrane deposition to determine the placement in the pseudo-time series.

Supplemental Figs

A table of contents has been added as the first page of the supplement

Figure S1:

- New; schematic of embryonic parasegments indicating where *ftz*, *eve*, *en*, and *slp1* are expressed and the corresponding body segments

Figure S2:

- New figure; schematic of *ftz* locus, highlighting the CREs in this study

Figure S3:

- Previously Figure S1; Transgenes of other *ftz* stripe-specific CREs
- Simplified the schematic in panel A

Figure S4:

- Previously Figure S2; schematic of CRISPR/Cas9 scheme for *ftzΔ2* genome edit
- No changes made

Figure S5:

- Previously Figure S3; schematic of CRISPR/Cas9 scheme for *ftz2ΔFlanking* genome edit

- No changes made

Figure S6:

- Previously Figure S4; schematic of *ftzΔUPSΔZ* double deletion mutant
- No changes made

Figure S7:

- New figure; boxplots of *ftz* HCR data: fluorescent intensity and width of *ftz* stripes in central 10% of wild type, *ftzΔ2*, and *ftz2ΔFlanking* embryos near end of stage 5 (cellular blastoderm)

Figure S8:

- Previously Figure S5; additional cuticle preparations from *ftzΔUPSΔZ* samples
- No changes made

Figure S9:

- Previously Figure S6; schematic of *eveΔ2*

Figure S10:

- New figure; boxplots of *eve* HCR data: fluorescent intensity and width of *eve* stripes in central 10% of wild type and *eveΔ2* embryos at both early stage 5 and late stage 5

Figure S11:

- New figure; *slp1* chromogenic in situ stains in wildtype and *eveΔ2* embryos, originally in Figure 6. Moved to this new figure to make space for the new *engrailed* in situs from the *eveΔ2/eve3* transheterozygotes.

Figure S12:

- New figure; additional *engrailed* chromogenic in situ stains to represent the variation in *engrailed* expression in these mutants

Supplementary Tables

Table S1

- New; cis-regulatory elements used in this study with some information to guide readers

Table S2

- New; HCR analysis of *ftz* stripe expression in wildtype, *ftzΔ2*, and *ftz2ΔFlanking*. Color-coded to match Fig. S7

Table S3

- New; HCR analysis of *eve* stripe expression in wildtype and *eveΔ2*. Color-coded to match Fig. S10

Table S4

- New; viability assay of *ftz* and *eve* stripe 2 alleles when reared at room temperature vs elevated temperature.

Table S5

- New; viability assay of *ftz* stripe 2 alleles when transheterozygous for either a hypomorphic or null *ftz* allele
- Table S6New; viability assay of *eve* Δ 2 allele when transheterozygous for either a hypomorphic or null *eve* allele.

Table S7

- Previously Table S1; list of primers. Unchanged except for nomenclature (for example, the change of *ftz*2 Δ 2+7 into *ftz*2 Δ Flanking)

Table S8

- Previously Table S2; sequences of regulatory elements and in situ probes used in this study. Unchanged except for nomenclature.

Fischer et al. Response to reviews

Reviewer 1: SUMMARY OF THE ADVANCE MADE IN THIS PAPER AND ITS POTENTIAL SIGNIFICANCE TO THE FIELD

In this nice study the authors rigorously dissect the role that individual pair rule stripe enhancers play in specifying the body plan of the *Drosophila* embryo. Focusing first on *ftz* they identify its stripe 2 enhancer, and then carefully test its function by deleting it in the context of the endogenous locus using CRISPR/Cas9. They show that even though this alters the early expression of *ftz*, the expression of key *ftz* target genes remain essentially unchanged. In addition, under lab conditions the flies are viable and show no significant defects in segmentation. From previous work it was known that two other 7 stripe enhancers also drive expression in the stripe 2 domain (UPS & Z) and so using a similar approach the authors simultaneously deleted these and assessed the impact this has on patterning. They showed that while this compromised viability, and lead to defects in segments associated with other stripes, the stripe 2 enhancer on its own is sufficient to specify correct target gene expression around it and direct proper segmental fate. To test the generalizability of their findings, they turned to *eve* and deleted the famous *eve* stripe 2 enhancer from the endogenous locus. Surprisingly, removing this enhancer has no discernable impact on the expression of *eve* targets, segmentation, or viability. This is even though the deletion causes a clear loss of *eve* expression in the stripe 2 domain until quite late in *nc14*.

These results are significant, novel and likely to be of broad interest to the developmental biology community, given that they challenge the conventional wisdom of how one of the model patterning systems in development works. The paper is well written, and the data reported clearly justify the conclusions drawn.

We thank the reviewer for noting the care we have taken in this study and for recognizing the significance and likely broad interest this study will raise in the community.

SUGGESTIONS TO AUTHORS

The only major issue I have is that previous studies where the *eve* stripe 2 enhancer was deleted in a rescue context are not cited or discussed (Ludwig M. Z., et. al., 2005 PLoS Biology & 2011 PLoS Genetics). In contrast to this work, these past studies found that the enhancer was essential for viability in the context of a transgene capable of rescuing *eve* null mutant flies to fertile adulthood. However, this apparent discrepancy can be reconciled by the fact that their unmodified transgene behaved like a hypomorph, showing reduced viability, especially as a hemizygote.

Thank you. We certainly should have cited the Ludwig papers. We have now added citations to their results in the Results, Discussion and legend to Figure 6. We agree with your explanation of the apparent discrepancy being a quantitative difference and the crosses to *eve* alleles that you suggested support this (see specific responses below and new text in Results and Discussion).

While I don't think this is necessary for publication, I do believe the authors have an opportunity to significantly extend the scope of the study with some straightforward additional experiments. In my opinion these have the potential to elevate this study from being a nice paper to a textbook example.

The authors already show that the removal of the stripe 2 enhancers of both *ftz* and *eve* are phenotypically buffered for flies kept under ideal lab conditions. Yet both enhancers are deeply conserved, which begs the question are they playing a functional role or are they merely vestigial? In the past several decades multiple examples in fly (*sna*, *svb*, etc.) and mouse (*gli3*, *shox2*, etc.) development have come to light where seemingly redundant enhancers (with expression patterns that partially overlap in space and/or time) have been shown to be required for canalization. In the absence of environmental or genetic stresses individuals missing single enhancers typically showed no phenotypes, similarly to here. Yet, as soon as they are subjected to stress individuals missing enhancers show severe phenotypes, while those with both are typically fine.

To test if either *ftz* stripe 2 or *eve* stripe 2 are required for canalization the authors would simply need to raise their flies at higher temperatures and see if they start to see denticle belt defects. A genetic stress could be applied by crossing the enhancer mutants to *ftz* or *eve* nulls and observing if hemizygotes lacking the stripe 2 enhancers start to show segmentation defects. Definitely showing one of the most heavily studied enhancers in development is required for canalization, and that this is also true for other pair rule genes, would significantly extend the scope of the study.

We have done these experiments, which have delayed resubmission. They were definitely worth the time because the results are quite informative. These results showed, as the reviewer expected, that the stripe2 CREs contribute to robustness, even though they are not absolutely required for segments to form. Experiments reporting these findings have been added to results, discussion and have changed the overall "story" of the manuscript and, as such, are also mentioned in the Abstract and at the end of the Introduction (lines 55-57; 152-160). Detailed responses are below. These new results are so exciting to us - we are truly grateful for your suggestions.

1. Comparing viability of *eveD2* and *ftzD2* homozygotes to controls showed that even at standard rearing temperatures, their viability was less than wild type. At a higher temperature (28°C), the effect was even more pronounced. This shows, as the reviewer suspected, that although the homozygous deletion lines are viable and fertile, they are less fit than wild type. We have added a new Supplementary Table (Table S4) along with text in Results to report this (lines 420-431).
2. We have also applied the suggested genetic stress, crossing *eveD2* and *ftzD2* homozygotes to weak and strong alleles (new Supp Tables S5 and S6). This showed both stripe 2 alleles behave as weak hypomorphs that are lethal in combination with strong alleles, but with adult escaper transheterozygotes for weaker alleles. There are interesting differences that we report and which will be pursued in the future. For *eve*, we followed up in more detail to analyze the defects associated with lethality of the transheterozygotes. This revealed cuticle defects in the region of T1, which were marginally evident in the *eveD2* homozygotes but which were clearly detectable in the unviable transheterozygotes (*eveD2/eve³*, aka *eve^{R13}*). Further, expression of *engrailed* was perturbed, very similar to that seen by Ludwig et al., and exactly as predicted by the reviewer. These findings are reported in Results, new panels in Figure 6, a new supplementary figure (Fig. S12). New text has been added to the Discussion about these results, where we compare the Ludwig M. Z., et. al results to ours. (lines 445-481)

Given the decreased fitness, we deleted "fully" from "fully viable" on line 149

Minor Points: Thank you for these; we have made all the corrections, as listed below.

1) About midway down page 3 there is a sentence "... hour period after egg laying", this should be "... hour period after fertilization", since fly embryos can start developing before the eggs is deposited.

Corrected, also noted by reviewer 2, line 74.

2) At the top of page 13 the paragraph ends with "for regulation of Eve target genes". I think this should be more accurately stated as "late Eve target genes". Eve has other targets (e.g., ftz, odd & run) that it regulates before en and slp1 (which come on quite late) but how their expression might change is still unclear.

Corrected, line 418.

3) The first sentence in the Drosophila genetics section on page 19 could do with some rewording for clarity.

Simplified to avoid confusion, lines 703-704.

4) On page 20 it is stated that "Cuticle preparations used standard methods." This is too vague, either a clearer description is required, or an established protocol should be referenced.

We have added a detailed protocol in the methods section, lines 729-739.

5) Midway on page 20 "conjugated to Alex Fluor 647", should read "conjugated to Alexa Fluor 647"

Thanks for noticing this. Corrected, line 753.

Reviewer 2: SUMMARY OF THE ADVANCE MADE IN THIS PAPER AND ITS POTENTIAL SIGNIFICANCE TO THE FIELD

Pair-rule gene expression is a key model for transcriptional regulation, developmental patterning, and the evolution of development; in particular, the eve stripe 2 enhancer and the ftz zebra element are textbook examples that most developmental biologists will have passing familiarity with. However, since eve and ftz both have both stripe-specific and zebra-type enhancers, the regulation of these genes is more complicated than the textbook model, as pointed out by the lead author of this study 30 ago (Yu and Pick 1995).

In recent years it has become possible to study the function of specific enhancers by using CRISPR/Cas9 plus homology-directed repair to make precise genomic deletions, an advance over earlier approaches using transgenic reporter genes and (partial) rescue constructs. The Pick lab has previously investigated the effects of deleting the ftz zebra element (Graham et al 2021) and the ftz upstream element (Fischer et al 2024) individually. In this manuscript they delete the stripe 2 enhancer of ftz, the stripe 2 enhancer of eve, and make a double deletion of both ftz 7-stripe elements (zebra and UPS). The double deletion allows them to look at the function of the ftz stripe-specific enhancers in the absence of the zebra enhancers, while the stripe 2 enhancer deletions allow them to ask the reciprocal question for eve and ftz stripe 2.

These new fly lines enable a nice clean test of stripe-specific enhancer function vs zebra enhancer function. The stripe 2 enhancer deletions result in loss of early stripe 2 expression for both eve and ftz, while the 7-stripe double deletion results in loss of late ftz expression in all stripes. Thus, as expected from reporter studies, the enhancers drive expression in different temporal ranges, which overlap at late stage 5. Losing the ftz 7-stripe elements has stripe-specific effects on gene expression and downstream segment patterning which are anti-correlated with contributions from the stripe-specific elements (e.g. stripe 4 which has no stripe-specific element is worst affected, but several of the other stripes/segments are fine); this is also as expected. Loss of the ftz stripe 2 enhancer produces no apparent effects on late ftz expression or downstream segmentation; this is not necessarily expected (why then is the ftz stripe 2 enhancer present - perhaps it provides patterning robustness under less ideal environmental conditions?) but it is consistent with the known roles of Ftz in the segmentation network, given that Ftz is not important for early patterning of other pair-rule genes, and its key targets (e.g. various segment-polarity genes) are not expressed until late stage 5.

The headline result of the study, however, is that the eve stripe 2 enhancer is dispensable for viability and normal segmentation (though does seem to contribute to patterning robustness, given

the low frequency thoracic defects observed in the deletion mutants). This is really surprising! Not only because *eve* stripe 2 is one of the most famous and best studied developmental enhancers, but because the early *eve* stripes have previously been shown to play important roles in pair-rule gene cross-regulation. Indeed these new results for *eve* stripe 2 contrast with earlier studies which found that the early expression from the *eve* stripe 4+6 element is required for normal segmentation (Fujioka et al 1995, Fujioka et al 2002). This work therefore suggests that the importance of the early *Eve* stripes differs from stripe to stripe, perhaps correlating with how the stripes of other primary pair-rules are regulated in the vicinity (more by stripe-specific elements or more by cross-regulation through 7-stripe elements).

A limitation of the study is that it only looks at the segmentation markers *en* and *slp*, and only at extended germband stage rather than late stage 5/stage 6 when they are first expressed; the immediate impact of the enhancer deletions on downstream segmentation gene expression is therefore unclear. However, the key finding is that both stripe 2 deletion lines give rise to viable adults with normal segments, and this is clearly justified by the data shown.

SUGGESTIONS TO AUTHORS

My suggestions involve changes to the text and/or presentation of the figures.

Major comments:

1. Title/summary/abstract/discussion: the authors have chosen to highlight the functional redundancy of the *ftz* stripe 2 and 7-stripe elements and their expression overlap at late stage 5 as the take-away message of the paper. However, this finding for *ftz* stripe 2 does not generalise (e.g. it is not the case for *ftz* stripe 4 or 5) and in my opinion it also buries the lead result of the paper, which is that the *eve* and *ftz* stripe 2 enhancer deletions do perturb the early stripes, but with little apparent effect on subsequent development. The authors might consider revising/reframing the relevant parts of the text to make both sets of conclusions clear.

Thank you for this comment. We have totally rewritten the Abstract (lines 36-58) and also altered the end of the Introduction (lines 149-161), and added a new section to the Discussion (lines 564-595), to focus more on the *eve* result, including adding the results of the robustness experiments suggested by reviewer 1 (lines 420-481). We appreciate your point that experiments on the “famous” *eve2* CRE will likely be the most interesting aspect for many readers and have attempted to “unbury” it throughout. We do feel that the compensatory action of different types of CREs that respond to different types of regulatory cues/transcription factors will be the larger contribution of this paper in the long run and we suspect it will apply more broadly, including to other *ftz* and *eve* CREs, but also to other genes and types of genes. In any case, your suggestions/encouragement are very much appreciated.

2. Embryo staging within stage 5: the Methods states (L601) that “Cellularization substage within stage 5 of embryogenesis was determined by the degree of cellular membrane deposition as viewed from the eyepiece with brightfield settings”. It would be useful to have this staging information provided alongside the embryos shown in Figure 3 and Figure 7, similar to e.g. Figure 2 in Schroeder et al 2011. This information is included in the main text for the *eve* stripe 2 deletions (L340-352), but not for the *ftz* enhancer deletions. As such, it is not clear the extent to which the rows in Figure 3 represent equivalent developmental timepoints, nor how to relate Figure 3 to Figure 7. This is particularly important because Schroeder et al 2011 show the *eve* late element driving expression in stripe 2 much later than the *ftz* LacC element (30-40' vs 5-17' into stage 5). However, both Fig 3 and Fig 7 show the stripe 2 deletion lines activating stripe 2 in the 5th row out of 9, giving the impression they turn on at the same time. When exactly during stage 5 does each line activate stripe 2, and is this consistent with the reporter gene expression described in Schroeder et al 2011?

Very fair point. We have included staging information alongside the embryos shown in Figure 3 and Figure 7 by inserting photos of the plasma membrane progression into the upper right corner of each panel. An arrow indicates the apical-to-basal progression.

I would be reluctant to directly compare the timing of expression of reporter transgenes to the endogenous gene, especially if we are talking about a few minutes in time. We know that transgene expression is influenced by the site at which insertion in the genome occurs, and these effects can be very large. Thus, any conclusion about CRE function through this type of comparison would be confounded by the random or selected site of insertion, and I am afraid we risk going down a rabbit hole in pursuit of a question we would not be able to answer.

3. Existing literature: a very pertinent previous study should be referenced and discussed: Fujioka et al (2002) *Development* 129, 4411-4421 - see in particular Figure 3. Fujioka et al separately deleted the eve 4+6 enhancer and the eve late element from an eve rescue transgene in an eve null background, allowing them to assess how these stripe-specific and 7-stripe elements contribute to patterning. (See also the earlier experiments investigating early vs late Eve function in Fujioka et al 1995.) For the eve 4+6 deletion, the thin late stripes still appeared in stripes 4 and 6 despite the loss of the early stripes, similar to the stripe 2 deletion described here. However, the late stripes only partially rescued segmentation and the resulting flies, although viable, had two fewer abdominal segments. (The earlier work in Fujioka 1995 suggests that the segmentation defects are mediated by misexpression of genes that would normally be repressed by Eve, such as runt, prd and slp.) The different result found here for the eve stripe 2 deletion is very interesting; the stripe-specific outcomes of these SSE deletions should be highlighted and discussed. For the late element deletion, late eve expression was lost as expected. Some of the lines rescued ~10% of individuals to adulthood, but there was a derepression of slp into the cells that would normally express late eve, showing the importance of this element for stabilising the parasegment boundaries.

Thank you and I am embarrassed to have missed this Fujioka et al. stripe 4+6 deletion result, especially for citation in our previous paper on the *ftz* zebra element deletion. We have now referenced the Fujioka papers throughout the manuscript but focused more on the direct comparisons to the papers by Ludwig et al., that looked specifically at eve stripe 2 and are very relevant to the results we show here. We have added text in results and quite a long section in the Discussion to address differences found by their approach and ours (lines 471, 574-575). A broader discussion and comparison of these types of experiments is warranted; perhaps a review article is in order? We do note, as suggested by reviewer 1, that two copies of these eve transgenes were required for decent rescue and, as was the case for *ftz* rescue (Hiromi and Gehring 1985), rescue is never complete, for reasons that have never been pursued.

Additional minor comments:

Title: consider word choice. "divergent" has evolutionary connotations and might give the misleading impression that the stripe-specific and 7-stripe enhancers are homologous to each other. Similarly, "compensatory" might suggest that one enhancer upregulates or downregulates expression depending on the presence of the other enhancer.

We changed this to "different types" but have retained "compensatory" since even though they are different, they compensate for each other in that segments are formed and development proceeds.

L46: "the classic eve stripe 2 CRE is not necessary for the expression of eve stripe 2" - this is only true for the late expression, and even this seems to be thinner than the wild-type expression (see comment on L362). "...or for Eve's regulation of downstream target genes": we do not know this to be the case for the early regulation for en and slp, nor for earlier expressed target genes such as odd and prd.

Yes, thank you. We have now re-written the Abstract and removed that phrase (lines 52-53).

L64: "In their absence, segments do not form." - for most pair-rule gene mutants, segments still form, just only half of them.

Added "corresponding" before "segments do not form" (line 73).

L65: "arising and then rapidly fading during an ~2 hour period after egg laying." - 2 hrs AEL would be the beginning of stage 5, before stripe formation.

Changed this to - 2.5 hours after fertilization, also noted by reviewer 1 (line 74)

L96: "A Late-acting (L) autoregulatory 7-stripe CRE was also identified for eve" - subsequent studies argue that this enhancer is only indirectly autoregulatory, i.e. repressed by genes that are themselves repressed by Eve: see e.g. Manoukian and Krause, Genes and Development (1992) p.1748, section "Regulation of the en and eve genes".

Good point. We deleted "autoregulatory" as the mechanism underlying this enhancer activity is not relevant to our study (line 106) and added the reference to Manoukian and Krause.

L113: "odd-skipped (odd) and runt (run) contain stripe-specific elements for all stripes" - the paper cited (Schroeder et al 2011) actually says that odd lacks SSEs for stripes 2, 4, and 7.

Thank you for pointing out this mistake. We deleted "for all stripes" (line 124-125)

L114: "sloppy-paired (slp) and paired (prd) are regulated in a stripe-specific fashion for stripe 1, but only 7-stripe CREs have been identified for the remaining stripes" - prd has a SSE covering both stripes 1 and 2 (Ochoa-Espinosa et al 2005) and also has regulatory sequences specifically associated with the posterior part of stripe 2 (Gutjahr et al 1993).

Thank you for this correction and references. We changed this to "for only anterior stripes" and added the Ochoa-Espinosa reference (line 126-128).

L130: "These studies" - unclear which studies are being referred to as only Graham et al 2021 is cited in the previous sentence.

Changed to "this" line 141

L153 (and other instances): "ftz-6" - Schroeder et al 2011 used the slightly different name ftz₍₋₆₎.

Replaced throughout (3 changes at lines 168, 621, and 1025; removed lines 175-177 and 193-195).

L173: "ftz-6R>lacZ was expressed in a strong stripe overlapping ftz stripe 2 and a very weak (near background levels) stripe overlapping ftz stripe 7" - for ftz₍₋₆₎, Schroeder et al 2011 show stripe 7 appearing considerably later (phase 3) than stripe 2 (phase 1), but the late stripe 7 doesn't look particularly weak. Was stripe 7 very weak even in older embryos?

Yes, this was indeed a bit variable in our experiments, but stripe 7 was always weaker than stripe 2. For example, in this image we pasted above, stripe 7 looks stronger. We changed this to "expression was observed... faintly in stripe 7" in the text and added the Schroeder reference to this sentence (line 184).

L176: "The weaker expression of stripe 2 for the ftz2+7 transgene compared to the ftz2 transgene" - it's unclear from the methods how this quantitative assessment was made. E.g. were staining reactions run simultaneously for both reporter lines and stopped after a standard time?

This difference is clear in the HCR staining in Figure 1, which clearly shows a stronger, wider signal in the ftz2 transgene than in the ftz2+7 transgene. We have added that to the text ((compare Figs. 1C to 1F), lines 195-196).

L186: "A second deletion was generated to test whether the regions flanking ftz2 that are present in ftz2+7..." - without studying Fig S3 it's unclear from this description that the deletion removes the just the flanking regions while retaining the ftz2 enhancer, so this could be stated more explicitly in the main text.

Yes, we agree, this was/is confusing. We have added text to make this more explicit (lines 207-210) and have changed the name of the deletion from *ftz2Δ2+7* to *ftz2ΔFlanking*, which hopefully will improve clarity - at least others in our lab found it to be helpful. We have also added a new Supplementary Figure (Fig. S2) showing the *ftz* locus and each of our *ftz* genome edits schematically, and a Supplementary Table to summarize all the CREs (Table S1).

L199: "we asked if: 1) the combined input of these 7-stripe CREs is necessary for segmentation and 2) stripe-specific CREs are sufficient to establish segmentation without any 7-stripe CRE contribution" - the distinction between these questions is not clear to me.

We changed this to "we asked if the combined input of these 7-stripe CREs is necessary for segmentation or if stripe-specific CREs are sufficient to establish segmentation without any 7-stripe CRE contribution" and removed the two separate questions (lines 225-227)

L214: "for most *ftz* stripes, stripe-specific CREs can provide sufficient levels of *ftz* for wildtype-like function." - and presumably a sufficient spatial refinement of the stripe as well.

Thank you. We added "and spatial refinement" to the sentence (line 244)

L231: "peak expression is reached for all seven *ftz* stripes" - was this measured? Include details in methods.

While we have quantified the level of *ftz* transcript using fluorescent in situ HCR at the end of stage 5 in the new results here (lines 269-272; Fig. S7, Table S2) and in our previous work (Graham et al., 2021; Fischer et al., 2024), we have not quantified past this time point. This is largely because our previous work suggests that substantial loss of transcription after this point has a minimal impact on segmentation and proper development. We also believe this point has been made many times in the literature and also in our observations of stained embryos.

L269: "At the next time point" - see major comment 2 re embryo staging.

As per that comment, we have added the staging information to the figure.

L279: missing word: "during the blastoderm [stage]"

Added, line 311-312.

L287: "reduced but still detectable in either *ftzΔZ* or *ftzΔUPS7S* homozygotes" - add references to this previous work.

Added, lines 320-321.

L302: "It appears that the proper development of *Drosophila* is refractory to losing a stripe-specific CRE." - unclear as written whether this is meant to be a general statement or refers only to stripe 2. See also major comment 3.

We removed this sentence (lines 336-337).

L306-328: It is not clear to me from the text how often defects were found in regions corresponding to *ftz* stripes 2, 3, and 6. E.g. line 315: "Though the exact expression pattern varied between embryos" - does this include defects related to stripes 2/3/6 or only 4/5/7? If the former, the conclusion that "ftz stripe-specific CREs alone are sufficient for consistently directing segmental fate for T2, A1, and A7" needs to be qualified, due to the loss of patterning robustness observed. There do seem to be *slp* and/or *en* abnormalities around parasegment boundaries 4/6/12 in Fig 5D/E/I/J. Note also that changes to the duration or spatial distribution of *ftz* expression may be contributing to the segmentation phenotypes, in addition to the expression level changes discussed.

Thank you for this thought-provoking point. Since we wanted to highlight the defects from *ftz* stripes 4, 5, and 7, we have re-arranged this sentence to flow more gracefully to the next sentence, which touches upon the weak *en* expression in other even-numbered parasegments (lines

349-350). For the *slp1* abnormalities, we observed defects in these other even-numbered parasegments ~1/3 the rate that we saw defects to *slp1* stripes 8 and 10. We have removed lines 360-363 to qualify and soften the conclusion highlighted.

L310: "this corresponds to decreases in levels of ftz stripes 4 and 5" - it would be helpful for readers to have a reference diagram relating the locations of the ftz stripes to those of the en/*slp* stripes and the eventual segments of the larval cuticle.

Very helpful suggestion! We have made a new Supplementary Figure (Fig. S1) to illustrate this and clarify the alignments. It is cited throughout the text.

L312: "slight expansion of individual stripes" - this points to a more general role for the ftz 7-stripe elements in maintaining the stability of the parasegment boundaries, as found for the eve late element (see major comment 3 above).

Yes, agreed, but since this is a well-established role of the 7-stripe CREs, we have not added commentary at this point in the results section.

L326: "Thus, it appears that ftz stripe-specific CREs alone are sufficient for consistently directing segmental fate for T2, A1, and A7. However, A3 fails to form (there is no stripe 4 CRE), and A4 is consistently fused to A5." - again, it would be useful for the reader to have a reference diagram relating the Ftz stripes to the eventual segmental boundaries.

As per above, we have added new Supplementary Figure S1. Again, very helpful suggestion.

L338: "Minor defects in the thorax were observed in some cases" - this suggests that the early stripe might increase the robustness of patterning, even if most embryos can do without it.

Yes, agreed but these were too subtle to diagnose. The additional experiments suggested by reviewer 1, specifically examination of transheterozygotes, has now addressed this and allowed us to reveal the defects. We have added new Fig. 6E to show examples of cuticle defects in the transheterozygotes and describe this in Results (line 463-472).

L341: "Like ftz, establishment of the eve stripes occurs in distinct phases" - with the caveat that Surkova et al 2008 found that (protein) expression patterns of both genes varied somewhat within timeclasses, and the order of stripe emergence also showed some variation.

True enough. We added "albeit with slight variations within each time class" to the sentence and cited the Surkova paper (line 378)

L360: "early stripe 7 seemed to be weaker than wildtype" - longer versions of the eve stripe 2 enhancer drive expression in stripe 7 as well [see e.g. Staller et al, PNAS (2015) and references therein], consistent with this observation.

Thank you for this reference and information. Upon analysis and quantification of in situ HCR images (new Supplementary Figure S10 and Table S3), it is apparent that there is not a statistically significant difference in eve stripe 7 intensity comparing wildtype and the stripe 2 CRE deletion mutant. Accordingly, we have removed this sentence (lines 396-398).

L362: "and stripe 2 appeared as a full, clear stripe (asterisk, Fig. 70)" - it's difficult to judge conclusively from the colorimetric in situ, but this stripe appears to be narrower than the wildtype stripe 2 in 7F, as would be expected from the known expression pattern of the eve late element relative to the SSEs.

Surprisingly, our analyses of eve stripe width from our in situ HCR images suggests that there is no statistically significant difference of stripe 2 width comparing the wildtype and stripe 2 deletion mutant embryos. This is true for both early stage 5 and late stage 5 embryos, and it can be seen in our new Supplementary S10 and Table S3. This information was added to the text (lines 405-410).

L368: "Despite this, downstream target gene expression was not disrupted in *eveΔ2* mutants." - see comment above on L46.

Added "late" to this sentence (line 411-412).

L371: "in *eveΔ2* mutants, expression of *slp1* and *en* were both indistinguishable from wildtype" - it's hard to judge from the colorimetric in situ, but it looks possible that the spacing between stripes 3 and 4 is narrower than in wild-type.

You may have a keener eye for this than we do. We have changed this to "appeared wildtype-like" (line 415) and added "although we cannot rule out small variations in stripe spacing or quantitative differences in expression levels of individual stripes" (lines 415-416). Your observation is in keeping with our results for the *eve* transheterozygotes, but the patterns do look quite wildtype-like to our eye.

L373: "or for regulation of *Eve* target genes." - see comment on L46.

Added "late" (line 412 and 418)

L384: "which respond to the same regulatory cues as primary enhancers" - I don't think this is generally how the terminology is understood - see e.g. Staller et al, PNAS (2015) for a counterexample. The key criterion would be overlapping expression, not similar regulation.

We believe that this is generally the implied meaning of the term and the way it is understood by readers. We have cited references that explicitly state this.

L386: "the 7-stripe and stripe-specific CREs 'get to the same finish line but by different routes.'" - however their expression patterns only overlap transiently in time, and for the *eve* late element the spatial overlap with the SSEs is only partial.

We clarified that what we mean by "finish line" is "segmentation" (lines 493-494). Since the different types of enhancers can compensate for each other, despite their differences in spatial and temporal expression, enough *eve* expressed in the pattern required to activate downstream genes and promote segment formation.

L394: "Hairy represses all *ftz* stripes after they are established" - *ftz* derepression begins quite early in hairy mutants; I'm not sure all the *ftz* stripes are established beforehand.

Yu and Pick (1995) make a pretty good case for the quoted statement.

L395: "In the case of autoregulation, 7-stripe CREs respond to activators" - see comment above for L96.

We edited this sentence to only talk about *ftz*, for which direct autoregulation has been extensively documented. It now reads "In the case of direct autoregulation, best documented for *ftz*, the 7-stripe CRE (UPS) responds to activators: *Ftz* and *Ftz-F1* activate all *ftz* stripes through the UPS." (lines 503-504).

L397: "The existence of these two different routes to stripe expression was thought to fall neatly into a two-tier system, with stripe-specific CREs establishing stripes and 7-stripe CREs refining them." - add references to illustrate this.

We based this statement on our own paper from 1995 that summarized this view in light of existing literature at the time, along with our own data. However, in shortening the text to meet the 7000 word limit, we have removed that sentence.

L397-403: "...The fact that stripe 2 reaches wildtype levels for both *ftz* and *eve* after deletion of the respective stripe 2 CRE shows that 7-stripe CREs can establish stripes independently of the earlier action of stripe-specific CREs." - given that secondary pair-rule genes lack SSEs for most

stripes and primary pair-rule genes can lack SSEs for specific stripes (e.g. *ftz* 4, odd 2, 4, 7), this was already known.

We don't really agree on this because the fact the SSEs have not been identified for particular stripes does not mean that they do not exist. We choose to leave this statement as written. Even if it was already "known" the statement is not incorrect as written - we are not saying it's a new finding, just that our results show it.

L404: "begs the question of how these two types of regulatory interactions arose and came to compensate for each other during evolution." - this has been discussed in the context of the evolution of long-germ segmentation from an ancestral short-germ state. See for example Fig 2 in Peel and Akam 2003 or Fig 5 in Clark, Peel and Akam 2019.

Yes, we should have cited these papers, but these questions still do logically and directly follow from our findings. We added "see discussion of these questions in Peel and Akam (2003) and Clark et al. (2019)" to lines 513-514. Note that our own work on *Oncopeltus* also raises these questions but we have not cited it here because we feel that our findings in this paper raise these questions on their own.

L414: "downstream genes are not activated until a certain time" - the role of Opa in activating late segmentation gene expression is relevant here (see e.g. Clark and Akam 2016, Soluri et al 2020, Koromila et al 2020).

During our edits, we had originally added these references. However, to cut down our word count, we ultimately deleted this section of the Discussion (lines 516-537).

L416: "peak-level expression in stripes at the end of cellularization/start of gastrulation (end stage 5/beginning stage 6) is crucial. Their expression earlier (as shown by stripe-specific CRE deletions) or later (as shown by the UPS deletion) is of no consequence" - this might be true for *ftz* (though see comment for L312 above) but it is not generally true for other primary pair-rule genes. *eve* and hairy mutants both show significantly perturbed downstream gene expression at mid stage 5, for example. See also major comment 3 above.

We have removed this part of the Discussion to reach our word count and focus on the *eve* stripe 2 CRE deletion (lines 516-537). We do note that the new experiments on robustness further clarify the role of the stripe2 CREs. We have also added discussion of the Fujioka and Ludwig results to a new section in the Discussion (lines 564-595).

L437-440: The meaning of "provides enough regulatory information" is ambiguous; the intended meaning seems to be "drives enough expression".

We have removed this section of the discussion to have more words to describe the role of stripe-specific CREs on robustness (lines 539-562; new section 564-595).

L457: "it has an autoregulatory role" - see comment above for L96.

We have removed this entire section of the Discussion and replaced it with a new section, "Stripe 2 CREs contribute to robustness." Based on the new experiments we have added to the revision.

L459: "although this remains to be tested, it is likely that the *eve* Late CRE is as dispensable as the *ftz* UPS" - see major comment 3.

As per above, this section of the Discussion was removed. However, just for the sake of conversation, given how our results compare to the rescue results of the Ludwig et al papers on the *eve* stripe 2 CRE, we stand by this speculative comment. We shall see.

L600: "Images were taken at a resolution of 1024 px²." - 1024 x 1024 px.

Fixed (line 758)

L861: typo: "partial f A4/A5 usion"

Fixed (line 1043)

L884: "Brightfield images of ftz expression time series throughout stage 6 (gastrulation)" - the embryos in rows 2-4 seem to be at stage 7.

Thank you for catching this - we have corrected the legend accordingly (line 1069).

L901: "(D) Wildtype cuticles" - typo, should be (C).

Fixed (line 1086-1087).

L904: "Downstream gene expression is not perturbed in eve Δ 2 mutants" - see comment for L371.

We removed "is not perturbed" (line 1089).

L917: "(K) A weak stripe 7 start to appear along." - part of sentence missing?

We have substantially simplified the figure legends, including the removal of lines 1103-1111.

Figure 1

- It could be clearer on the diagram that ftz2 is a renamed 413 and ftz2+7 is a renamed ftz-6.

We agree - the history of the names is very tedious. We have tried our best in the text, but have added a new Supplementary Figure (Fig. S2) and Supplementary Table (Table S1) to define the CREs more clearly. We have also simplified the diagram in panel A of Figure 1.

- Consider colour-blind friendly colour combinations for the merges in D and G. Colors were changed. Thank you for the nudging on this.

- Labelling the eve and ftz channels on the figure (not just in the legend) would be easier for readers. Done.

Figure 3

- See major comment 2. As per our responses above, we added insets showing the progression of membrane invagination to each panel.

- Adding annotations to number emerging stripes might be helpful. Done.

- There is some variation in embryo orientation which could be noted in the legend. Added, line 1048-1049 and 1103.

- The ventrolateral embryo in CG should be flipped vertically. Thank you for indicating. We have gone through the panels and re-oriented some in Figure 3 and Figure 7, including the one in the current CF (originally CG).

Figure 7

- See major comment 2. As per our responses above, we added insets showing the progression of membrane invagination to each panel.

- As with Fig 3, there is some variation in embryo orientation which could be noted in the legend.

- Some embryos should be vertically flipped, e.g. M. Thank you. We have gone through the panels and re-oriented some embryos.
- Green / magenta would be more colourblind-friendly than green / red. Done.
- eve and ftz channels could be labelled on the figure. Done.

Supplementary figures:

- Add scale annotations for the ladders (S2-4, S6) Done.
- Fig S3, panel D: it is hard to relate the description in the legend ["405 bp fragment for negative events (lanes 6-7)"] to the figure because the obvious band in lanes 6-7 is for a shorter fragment; add additional annotations or explanation as appropriate. We had forgotten to write in the original manuscript that lane 6 is a negative genomic control and lane 7 is a no-template, water control. The shorter fragment is just primer. This has been added to the figure legend for Fig S3.
- Fig S4: state what ladder was used in legend. Add a sentence to the legend to explain why both bands are present (i.e. flies are heterozygous over balancer). The ladder sizes have been added. Matthew: Does he mean S6C, last 2 lanes? Does he just want us to say a sentence that they are heterozygotes? Please fix this. We have added the ladder to the legend and added a sentence explaining why both bands are present.

Reviewer: Erik Clark.

Reviewer 3: SUMMARY OF THE ADVANCE MADE IN THIS PAPER AND ITS POTENTIAL SIGNIFICANCE TO THE FIELD

This manuscript investigates the regulation of ftz expression in the fly embryo and the functional consequence of enhancer deletions. The authors compare two types of enhancer deletions: stripe specific CRE mutants (delta stripe 2 or delta stripe 2+7) with a larger deletion of two entire regulatory regions UPS and Z, known to operate as two separate 7-stripe CREs. Through mostly fixed approaches, they investigate the timing, location and levels of ftz expression (as well as Ftz targets) in these various enhancer deletions. They also perform a phenotypic analysis to examine the functional consequences of enhancer deletions.

Finally, the authors compare ftz regulation to that of eve. Indeed, eve is supposed to rely on stripe specific regulations while ftz was thought to be regulated by a two 7-stripe CREs. Surprisingly, they found that deletion of eve stripe 2 enhancer has no functional impact on embryogenesis. As observed for ftz, the deletion of eve stripe 2 enhancer initially perturbs eve expression in this domain, but there are compensatory effects such that by the end of nc14, eve expression appears normal.

The main conclusion of this work is that 7-stripe CREs can compensate for the loss of a stripe a specific enhancer, even if these 2 types of enhancers are regulated by different input. They propose a model in which a periodic pattern can be established by two distinct manners (1 stripe at a time that is unprecise then sharpened or 7 stripes regulated by TF which are themselves expressed in a periodic manner). These two types of regulations can compensate for each other, thus providing redundancy to the system.

Overall, the manuscript investigates an original question regarding spatio-temporal regulation of transcription by enhancers and the regulation of ftz stripes along the fly embryos A/P axis is the best system to address this question. I appreciate the systematic attempt of the authors to link gene expression levels to functional output with phenotypic analyses. My major comment is that the questions of transcription dynamics (levels, timing and spatial coordinates) require a more quantitative approach than colorimetric in situ hybridization. It seems difficult to apprehend the questions asked without subcellular resolution of mRNA production and without a dynamic approach. Moreover, I found the manuscript (and figures) difficult to read.

SUGGESTIONS TO AUTHORS

Major Comments

Figure 1:

I assume colorimetric ISH are with a lacZ probe. This should be mentioned.

We added “using a *lacZ* probe” to the Legend to Figure 1 (line 1027). This is also mentioned in Methods.

Why is the *ftz2* Reverse transgene more efficient than the forward in driving expression in stripe 2? (Colorimetric signal in B versus C)

We do not know the reason for this but suspect the presence of an insulator or other element influencing chromatin structure is involved. Pursuing this experimentally is outside of the scope of the current manuscript. We note that this reverse orientation effect was first seen by Hiromi et al (1987) and Pick (1990).

Ftz6 was renamed *Ftz2+7*, but the expression in stripe 7 is very weak: it can only be observed with the colorimetric ISH, and only in the reverse position. Can the author comment on this?

Yes, we agree and also cite the original paper from Schroeder et al 2011 on this point (lines 184-185). We presume that their publication also examined a reverse orientation transgene, but this was not clear enough in their Methods for us to cite. As for the previous comment on the reverse orientation effect for *ftz2*, pursuing chromatin elements that may be underlying this is outside of the scope and direction of this manuscript.

Figure 3 reconstitutes the timing of appearance of *ftz* expression in each stripe, in various enhancer deletion contexts. Nowadays, this temporal question is generally examined with live imaging, with MS2 reporters, a technique widely employed to study gene expression dynamics in the early fly embryo.

Indeed, colorimetric ISH is not quantitative, yet the description and interpretation of the entire figure 3 relies on expression levels. If live imaging is challenging to implement for this lab, a FISH (ideally smFISH now inexpensive with techniques as smiFISH, for example Calvo et al, <https://doi.org/10.1038/s42003-021-01803-0>) could be used, with *ftz* as well as an internal marker to better label the timing of *nc14* progression.

We are certainly aware of the live imaging techniques now used to study transcriptional dynamics in *Drosophila* embryos (note that we highlighted this approach in a recent review article Reding & Pick 2025). However, the study we present here is not aimed at detailing the types of transcriptional dynamics directed by the different cis-regulatory elements examined. That would be quite interesting to determine in the future, but it was not the goal of the current study. While these “nowadays used” techniques are powerful, not every technique is appropriate to every question. Rather, our primary focus was on the developmental effects of the deletions and the overall role of the different types of cis-regulatory elements in directing segment formation. We agree that colorimetric in situ is not a quantitative method for viewing gene expression and have modified language throughout the text. However, differences in level, location, and time of expression can be noted by staining batches of embryos simultaneously and paying careful attention to the stage of the embryos being compared. This approach also allows us to look at hundreds of embryos, rather than the small number that are typically used for higher resolution imaging. The types of changes we observed in gene expression are consistent with the developmental effects observed, such that this level of resolution, even if not rigorously quantitative, was sufficient for the questions we are asking in this manuscript.

We have added in an analysis of quantitative fluorescent in situ HCR (Figs S7 and S10, Tables S2 and S3) (Surkova et al., 2019; Choi et al., 2014). This was applied for *ftz* with the wildtype, *ftzΔ2*, and

ftz2ΔFlanking fly lines in addition to *eve* for the wildtype and *eveΔ2* fly lines. The results were consistent with our colorimetric in situ results, as shown in Figures 3 and 7.

We want to emphasize that we explicitly decided to limit the changes made to the genome in the deletion lines to avoid possible effects from other alterations to that could impact gene expression. That is, we wanted to ensure that our genome edits are minimally invasive - hence why we did not even put in a visual marker. We only removed the enhancer, and that's it - everything else is just the native *ftz* or *eve* genomic locus - no other changes made. Given the subtle effects seen for the stripe 2 element deletions on survival, which were especially revealed by the new experiments suggested by reviewer 1 (Tables S4, S5, and S6), adding tags to the mutant lines runs the risk of confounding the conclusion. Anecdotally, we have learned that insertion of MS2 sequences sometimes - although not always - results in homozygous lethality. Since we do not know if these insertions, even when viable, may have subtle effects on gene expression, we stand by our original idea of not utilizing tags in this particular study. It is interesting that Ludwig et al. (2005, 2011) made an *eve* stripe2 deletion that removed the same *eve* sequences (*eve* MSE) as ours, but we note they also inserted a *white+* gene as visible marker in its place. While our lines are homozygous viable, the Ludwig lines were homozygous lethal, suggesting that the insertion of *white*, along with its cis-regulatory sequences to allow expression and marker detection, into the native *eve* locus, itself disrupted *eve* expression. We believe that this comparison further justifies our approach of making simple deletions without inserting other sequence.

I assume that panels of Figure 3 are ordered according to nc14 progression (for example AA being younger than AD, again this nomenclature is confusing and not explained). However, without a labeling of the membrane invagination or an endogenous transcript, this temporal ordering remains challenging.

Yes, we added staging information to Figures 3. We explain that “The progression of *ftz* stripe expression was tracked temporally by following the progression of membrane invagination along the periphery to form a cellular blastoderm from a syncytium (see insets in each panel).” (line 376-377). We similarly revised Figure 7 (see below). Both of these figures have been revised to have insets added to show the progress of membrane invagination for each panel (see revised Figures 3 and 7). We also changed “identical to” to “indistinguishable from” in the following sentence “Despite these differences of *ftz* dynamics throughout the cellular blastoderm, *ftz* expression in both stripe-specific CRE mutants was indistinguishable from wildtype throughout gastrulation” (line 289).

-From Figure 3, the authors conclude that deleting *ftz2* enhancer or *ftz* 2+7 enhancer affects the early expression of stripe2 but this effect is somehow compensated by the end of nc14 (stage5). However, such a conclusion cannot be drawn without a rigorous quantification of stripe2 pattern. How precise are stripe 2 anterior and posterior borders? How comparable are TS intensities in the absence of *ftz2* or *ftz2+7* enhancers? Colorimetric ISH is not quantitative enough to conclude on transcription levels, timing nor spatial precision.

We have added Tables S2 and S3 in which we use fluorescent Hybridization Chain Reaction in situ to quantitate levels of gene expression, supporting our claim. We now state “This was validated by quantitation of gene expression levels by HCR, which found that the fluorescent intensity and width of *ftz* stripes in wildtype, *ftzΔ2*, and *ftz2ΔFlanking* late stage 5 embryos was statistically insignificant (Fig. S7, Table S2). Thus, early perturbations of *ftz* stripe establishment in stripe-specific CRE mutations can be compensated by seven-stripe CREs.” (lines 269-272) and “HCR confirmed the low levels of *eve* stripe 2 during early stage 5 compared to wild type to be statistically significant (Fig. 7U, arrow). This also confirmed and recovery to wild type levels during late stage 5 (Fig. 7V, arrow; Fig. S10, Table S3). The width of *eve* stripe 2 is similar in wildtype and *eveΔ2* mutants during both early stage 5 and late stage 5 embryos (Fig. S10). This suggests that like *ftz*, the late-acting, seven-stripe CRE of *eve* can compensate for stripe-specific CRE mutations,” (Lines 405-410).

-Figure 4 is the mirror Figure of 3 but examined at later stages (stage6). It shows a clear absence of *ftz* expression upon deletion of the UPS and Z control regions.

Yes, in the absence of both 7-stripe CREs, *ftz* expression faded during gastrulation. We state “As gastrulation began, *ftz* expression was nearly undetectable in stripes 1 and 5, and only dorsal expression of *ftz* was observed for stripes 2, 3, 5, and 6 (Fig. 4M). For the rest of gastrulation, *ftz* expression was undetectable (Fig. 4N-P), likely due to the absence of the UPS, which is necessary for late *ftz* expression (Fischer et al., 2024)” (lines 306-310).

-Figure 5: same comments as for Figure 3 and 4. The levels of expression could be more rigorously assessed with a more quantitative approach (smFISH or live imaging).

For the effect on downstream target gene expression, we are not reporting quantitative changes in the levels of gene expression but rather, spatiotemporal patterns of expression. To make this clear, we added, “although we cannot rule out small quantitative differences in expression level using colorimetric in situ hybridization” (lines 335-336) and deleted the following sentence. Note that we were able to detect changes in *en* and *slp* gene expression patterns in *ftzΔUPS⁷⁵ΔZ* mutants. We made similar changes to the text accompanying Figure 6 for *eve*, stating “although we cannot rule out small variations in stripe spacing or quantitative differences in expression levels of individual stripes” (lines 415-416). Also, although only for *eve*, the new data on *engrailed* expression in transheterozygotes bolsters our expectation that we are able to detect gene expression changes in our embryos (see Figure 6H-J and new Figure S10).

-Figure 6: as for Figure 3, side by side comparison can be achieved only of shown embryos are at the same stage. Indeed, *eve* expression is particularly dynamic in *nc14* (Bothma 2014). If embryos are of comparable stages, *eve* expression in stripe 2 appears perturbed not only at early stages but also in cellularized embryos: panel O compared to F. In fact, stripe 1 also appears thinner in O, compared to F. Panel P should be replaced.

We have replaced Panel P, as requested. As for Figure 3, we have added panels showing progression of the membrane to stage the embryos to Figure 7. By looking at the membrane in-set more closely, we determined that one of the rows in the original Figure 3 was superfluous, and it was removed. We believe the rows of Figure 3 and Figure 7 are now more properly aligned. By looking at these figures side-by-side, we determined that while *ftz* stripe 2 is evident in the *ftz* stripe 2 deletion mutants in row E of Figure 3 (Fig. 3 AE, BE, CE), *eve* stripe 2 is not evident until one row later (Fig. 7 F & O).

Minor Comments

Figures are difficult to read.

Some suggestions: split the colorimetric ISH and the fluo ISH into distinct panels with a clear white border.

The only figure that has both colorimetric and HCR in situ in the same panel is Figure 1. This figure is pretty simple, with only 2 HCR embryos shown. For all others, white borders separate all panels. For Figures 3 and 7, we have separated the colorimetric and HCR in situ with a white border.

The genotype/ name of transgene could be indicated above. The nomenclature *ftz-2F-hsp70:lacZ* is not easy to read.

We have made changes to the figure and also added Table S1 and Figure S1 to clarify the cis-regulatory elements used in our study.

Panel A: maybe no need to indicate the sizes? We are not sure what is meant here.

The manuscript is also not easy to read. I would avoid using too many acronyms in the abstract (PRG, CRE). The manuscript also contains many acronyms and jargon (UPS, Z, primary or secondary PRG, PR-stripe) and should be reformulated to be understood by a broad readership. It's unclear to me who are the primary and secondary PRG.

We apologize that the paper was difficult to read.

To clarify the primary and secondary assignments of pair-rule genes, we added gene names to the sentence summarizing early studies: “Early studies of PRG expression suggested that so-called primary PRGs, such as *eve* and *hairy*, are controlled directly by maternal and gap proteins while the 7-stripe patterns of so-called secondary PRGs, such as *ftz*, are controlled solely by pre-positioned primary PRGs” lines 81-83. Also, lines 122-126 describes the “newer” classification of these genes, with all gene names listed.

We changed the name of the deletion mutation that retained the core *ftz2* element but removed surrounding DNA from *ftz2Δ2+7* to “*ftzD2Flanking*.” We think this was the most confusing name and that it is now clearer.

For the names of the *ftz* cis-regulatory elements, we have added a Supplementary Figure (Fig S2) and Supplementary Table S1 to help with this but do not see a way or justification for changing the names of these elements that have been in the literature for many years.

Finally, for the abstract, we agree that acronyms are not appropriate and have removed them.

To understand which cis-regulatory information is kept upon deletion of the UPS and Z elements, these regions should be indicated in a summary figure of the *ftz* locus.

We agree and have added a new Supplementary Figure (Figure S2).

Second decision letter

MS ID#: dev.204872R1

MS Title: Compensatory action of different types of cis-regulatory elements buffers the striped expression of *Drosophila* pair-rule-genes

Authors: Matthew D. Fischer; Kristen Au; Patricia Graham; Leslie Pick; Minh Lê

Article Type: Research Article

Dear Dr Pick,

I have now received all the referees reports on the above manuscript, and have reached a decision. The referees' comments are appended below.

The overall evaluation is positive and we would like to publish a revised manuscript in Development. Referee 2 has a few remaining comments and suggestions for clarifying the text. Please attend to these comments in your revised manuscript and detail them in your point-by-point response. If you do not agree with any of their criticisms or suggestions explain clearly why this is so. Please send us a point-by-point response indicating your plans for addressing the referees' comments, and we will look over this and provide further guidance.

Reviewer 1

The authors have addressed all of my concerns. The multiple additional experiments they added substantially strengthened their original findings, and also significantly extend the scope of the study.

Reviewer 2

SUMMARY OF THE ADVANCE MADE IN THIS PAPER AND ITS POTENTIAL SIGNIFICANCE TO THE FIELD

The authors have engaged thoughtfully and thoroughly with the reviewer comments and significantly improved the manuscript. In particular, the new robustness experiments are very nice and reconcile various other pieces of the scientific picture. The reframed story offers an explanation for contrasting past results using rescue transgenes, explains why the apparently superfluous stripe 2 enhancers are present in the genome, and helps explain how enhancer contributions to segment patterning could have changed gradually over the course of insect evolution. The headline results, while still striking and surprising, now make much more sense.

SUGGESTIONS TO AUTHORS

I only have a few, very minor comments/suggestions:

L46 (Abstract): "Thus, stripe-specific elements contribute to robustness but are not absolutely required for segment formation." I suggest "Thus, these stripe-specific elements..." to limit the conclusion to what has been shown in the paper. It will be interesting to find out how far the result generalises to other SSEs.

L106: "thus explaining establishment of the periodic ftz pattern by gene(s) pre-positioned to impact all 7-stripes coordinately" Ambiguous sentence as currently worded; it is gene products that are pre-positioned rather than the genes themselves.

L121: "(except for stripe 1)" - this was changed to "anterior stripes" in L114 because of prd enhancers contributing to stripe 2; this later sentence should also be updated.

L128: It would be helpful to the reader to note somewhere around here that stripe 4 is thought to be the only ftz stripe that lacks an SSE, both for understanding the Z deletion results from Graham et al 2021 and for understanding the UPS Z double deletion results described later on.

L146: "further demonstrate that different types of CREs, which act at different time points". I suggest "act over different time ranges".

L212: "to the region specified by ftz stripe 3" presumably this should be "ftz stripe 4". Again, it would be helpful to the reader to more explicitly connect the discussion of the stripe 4-related patterning defects to the lack of an SSE for this stripe.

L457: "Hairy represses all ftz stripes after they are established (Howard and Ingham, 1986, Yu and Pick, 1995, Schroeder et al., 2011)." This is not critical to the manuscript and I'm not requesting a change to the text, but just to continue the conversation: we haven't seen embryos like Yu and Pick Fig3A in the data we have from hry mutants. The early ones seem to have fusions of 2-3, 4-5, and 6-7, similar to Schroeder et al 2011 Fig5. Maybe we are missing the right stages, maybe it is embryo-to-embryo variability, or maybe there is an allele difference (hry[7H94] is predicted to give a truncated protein).

Figure 6: legend text could clarify that the *evedelta2/eve3* genotype of the cuticles is presumptive (due to the segmentation defects).

Figure 7: reconcile figure and legend - legend currently says all panels except H are dorsal top, but some of the early embryos look to be dorsal rather than lateral orientation. Panels M and P look to be ventral top.

There are a few typos and other small textual issues remaining that can be caught by a careful copy edit.

Reviewer 3

The revised manuscript has been significantly improved. The message is now much clearer, better presented and quantified and conclusions are solidly supported by the data. I particularly liked the new data on robustness (with genetic or temperature stress).

I recommend publication and congratulate the authors for their work.

Second revision

Author response to reviewers' comments

We are pleased that reviewers 1 and 3 are now happy with the manuscript and appreciate the additional comments from reviewer 2 that will make further improvements to it. Responses below in blue.

Reviewer 1: The authors have addressed all of my concerns. The multiple additional experiments they added substantially strengthened their original findings, and also significantly extend the scope of the study.

Reviewer 2: SUMMARY OF THE ADVANCE MADE IN THIS PAPER AND ITS POTENTIAL SIGNIFICANCE TO THE FIELD

The authors have engaged thoughtfully and thoroughly with the reviewer comments and significantly improved the manuscript. In particular, the new robustness experiments are very nice and reconcile various other pieces of the scientific picture. The reframed story offers an explanation for contrasting past results using rescue transgenes, explains why the apparently superfluous stripe 2 enhancers are present in the genome, and helps explain how enhancer contributions to segment patterning could have changed gradually over the course of insect evolution. The headline results, while still striking and surprising, now make much more sense. Many thanks for your great ideas.

SUGGESTIONS TO AUTHORS

Thank you Erik for all of your suggestions, insights, and persistence in improving our manuscript. In the "highlighted" version, deletions are crossed out and added words are in blue. I would be happy to continue our broader conversation about segmentation "off line" in the future. Leslie

I only have a few, very minor comments/suggestions:

L46 (Abstract): "Thus, stripe-specific elements contribute to robustness but are not absolutely required for segment formation." I suggest "Thus, these stripe-specific elements..." to limit the conclusion to what has been shown in the paper. It will be interesting to find out how far the result generalises to other SSEs. Fair point! Done, and still within the 180 word limit.

L106: "thus explaining establishment of the periodic ftz pattern by gene(s) pre-positioned to impact all 7-stripes coordinately" Ambiguous sentence as currently worded; it is gene products that are pre-positioned rather than the genes themselves.

Another fair point. I tell this to my trainees all the time....Corrected.

L121: "(except for stripe 1)" - this was changed to "anterior stripes" in L114 because of prd enhancers contributing to stripe 2; this later sentence should also be updated. Done.

L128: It would be helpful to the reader to note somewhere around here that stripe 4 is thought to be the only ftz stripe that lacks an SSE, both for understanding the Z deletion results from Graham et al 2021 and for understanding the UPS Z double deletion results described later on. We added "for which no stripe-specific CRE has been identified" to this sentence.

L146: "further demonstrate that different types of CREs, which act at different time points". I suggest "act over different time ranges". Nice wording, done.

L212: "to the region specified by ftz stripe 3" presumably this should be "ftz stripe 4". Again, it would be helpful to the reader to more explicitly connect the discussion of the stripe 4-related patterning defects to the lack of an SSE for this stripe. Oh gosh, thank you. Corrected.

L457: "Hairy represses all ftz stripes after they are established (Howard and Ingham, 1986, Yu and Pick, 1995, Schroeder et al., 2011)." This is not critical to the manuscript and I'm not requesting a change to the text, but just to continue the conversation: we haven't seen embryos like Yu and Pick Fig3A in the data we have from hry mutants. The early ones seem to have fusions of 2-3, 4-5, and 6-7, similar to Schroeder et al 2011 Fig5. Maybe we are missing the right stages, maybe it is embryo-to-embryo variability, or maybe there is an allele difference (hry[7H94] is predicted to give a truncated protein).

This is interesting. I have re-read the older Ingham papers and see that Ingham and Gergen (1988) stated the same thing Yu and Pick saw "In h- embryos, the initial expression of ftz is established normally but the seven-stripped pattern fails to resolve properly (Fig. 3A)." This was using radioactive in situ's which were even less sensitive than methods used today but their figures are quite clean. I have added that reference. This is definitely something we are going to need to look into in more detail. One thing that has come to my attention recently is that the higher resolution imaging techniques such as HCR usually analyze a relatively small number of embryos, likely because of the time needed for each embryo to be processed. The colorimetric in situ technique is almost the opposite of this - we can view many, many embryos on a slide, even under a dissection scope. I have been thinking lately that even though the resolution - for example of your beautiful fluorescence doubles (figures that we have consulted often!) - is much higher at the cell level than the colorimetric staining, we do gain something from the larger number of embryos captured with the colorimetric technique.

Figure 6: legend text could clarify that the *evedelta2/eve3* genotype of the cuticles is presumptive (due to the segmentation defects). Yes, good point. We clarified this in the Figure 6 legend "(E) *eveD2/eve³* Larval cuticles of offspring from cross of *eveD2* homozygotes to *eve³/CyO*; left, wildtype-like, *presumptive eveD2/CyO*; right, anterior defects (arrowhead) and abdominal defect (asterisk), *presumptive eveD2/eve³*" and also added presumptive in other spots.

Figure 7: reconcile figure and legend - legend currently says all panels except H are dorsal top, but some of the early embryos look to be dorsal rather than lateral orientation. Panels M and P look to be ventral top.

As M and P are aligned now, the flatter edge indicative of the dorsal surface is indeed on top. When they are flipped, it becomes apparent that the current alignment is correct. We believe the alignment of the embryos is correct as-is.

There are a few typos and other small textual issues remaining that can be caught by a careful copy edit. Thank you, yes, we made a bunch of small changes/corrections.

Reviewer 3: The revised manuscript has been significantly improved. The message is now much clearer, better presented and quantified and conclusions are solidly supported by the data. I particularly liked the new data on robustness (with genetic or temperature stress).

I recommend publication and congratulate the authors for their work. Many thanks.

Third decision letter

MS ID#: dev.204872R2

MS Title: Compensatory action of different types of cis-regulatory elements buffers the striped expression of *Drosophila* pair-rule-genes

Authors: Matthew D. Fischer; Kristen Au; Patricia Graham; Leslie Pick; Minh Lê
Article Type: Research Article

Dear Dr Pick,

I am happy to tell you that your manuscript has been accepted for publication in Development, pending our standard publication integrity checks.